EMBO
Molecular Medicine

# circ-EGFR is a predictor of response to Cetuximab and a potential target in colorectal cancer

Silei Sui[1,2], Yuan Li [1,3], Joan Maurel [4] & Ajay Goel [1✉]

## Abstract

**Cetuximab, an EGFR-targeting monoclonal antibody, provides beneficial yet limited clinical improvement in *KRAS* wild-type metastatic colorectal cancer (mCRC). While circRNA dysregulation has been implicated in various cancers, the role of circ-EGFR in response to EGFR-targeted therapy in mCRC remains largely unexplored. Here, we identified circ-EGFR as a promising predictive biomarker for cetuximab response. Clinically, we first determined that tissue-based circ-EGFR biomarker effectively stratified responders from non-responders to cetuximab in mCRC, with an Area under the Curve (AUC) of 76.8%. Functional assays demonstrated that circ-EGFR enhances the sensitivity to cetuximab, whereas its depletion induces resistance in CRC. Mechanistically, we revealed that circ-EGFR functions as a sponge for miR-942-3p, resulting in the upregulation of GAS1, which activates the Hedgehog signaling pathway and promotes the efficacy of cetuximab in CRC. Importantly, we effectively translated this tissue-based biomarker into a liquid biopsy predictor for anti-EGFR response (AUC: 76.9%), highlighting its non-invasive potential. In conclusion, circ-EGFR is a significant predictor of cetuximab efficacy in mCRC, potentially aiding in patient selection and treatment management, especially for patients with low circ-EGFR expression.**

**Keywords** Cetuximab; circ-EGFR; Metastatic Colorectal Cancer; miR-942-3p; Therapeutic Response Prediction
**Subject Categories** Cancer; Digestive System; RNA Biology

See also: C Pilati & P Laurent-Puig

## Introduction

Approximately 50% of patients with colorectal cancer (CRC) develop metastasis during the disease course, but unfortunately, 20–25% of them already have metastatic colorectal cancer (mCRC) at initial diagnosis (Bando et al, 2023; Leporrier et al, 2006; Sung et al, 2021). Most patients diagnosed with mCRC are incurable, contributing to ~90% of CRC-associated mortality (Chaffer and Weinberg, 2011). From a biological standpoint, the epidermal growth factor receptor (EGFR) plays a pivotal role in regulating cell growth, migration, stromal invasion, and angiogenesis in many cancers, including CRC (Ciardiello and Tortora, 2001). Accordingly, anti-EGFR monoclonal antibodies, such as cetuximab, in combination with chemotherapy, have become the standard-of-care, first-line treatment options for mCRC patients with *KRAS* wild-type (WT) gene (Van Cutsem et al, 2016). Although the adoption of cetuximab has significantly improved clinical outcomes in patients with mCRC, its effectiveness is limited to only a small subset of patients. It has been estimated that, at best, the therapeutic efficacy of cetuximab is ~30–35% when combined with chemotherapy (Sorich et al, 2015). The limited therapeutic efficacy of cetuximab highlights our incomplete understanding of the mechanisms underlying resistance, emphasizing the need for clinically applicable predictive biomarkers in mCRC. Several studies have investigated potential tissue-based biomarkers for predicting progression-free survival (PFS) in mCRC, including tumor DNA mutations (Bhattacharya, 2023; Ohnmacht et al, 2023), Consensus molecular subtypes (CMS) (Stahler et al, 2023), and molecular alterations linked to the EGFR pathway (Battaglin et al, 2024; Bhattacharya, 2023). However, despite their potential benefits, these biomarkers have not been translated into clinical practice due to their insufficient accuracy. Moreover, cancer cells can regulate biological processes at the post-transcriptional level independently of DNA mutations (MUT). Thus, it is essential to explore novel biomarkers to enhance existing strategies for predicting anti-EGFR response and guiding precision therapy.

Circular RNAs (circRNAs) are a particular class of endogenous non-coding RNAs intricately formed through reverse splicing of linear RNAs to form a covalently closed loop structure (Li et al, 2018). Due to their unique structural configuration, circRNAs exhibit enhanced resistance to RNase-mediated degradation, are more stable, and possess a longer half-life (Li et al, 2020). Accumulating evidence suggests that circRNAs are abundantly expressed in tumors and play crucial roles in regulating tumor proliferation and metastasis in numerous human malignancies, including CRC (Chen, 2020; Kristensen et al, 2022; Wang et al, 2022). However, the functional role of circRNAs in mCRC and their ability to predict treatment response remain largely unexplored. Our study aims to fill this knowledge gap and explore the potential of circRNAs as predictive biomarkers in cancer treatment.

[1]Department of Molecular Diagnostics and Experimental Therapeutics, Beckman Research Institute of City of Hope, Biomedical Research Center, Monrovia, CA, USA. [2]Department of Oncology, The Second Affiliated Hospital of Dalian Medical University, Dalian, China. [3]Department of Clinical Laboratory, Yangpu Hospital, Tongji University School of Medicine, Shanghai, China. [4]Translational Genomics and Targeted Therapies in Solid Tumors, Medical Oncology Department, Hospital Clínic de Barcelona, C/C/ Villaroel, 170, 08036 Barcelona, Spain. ✉E-mail: ajgoel@coh.org

Circ-EGFR, originating from EGFR, is one of the estrogen-signaling-related circRNAs. Previous studies have shown that circ-EGFR has dual effects on tumor initiation and progression in various cancers. It promotes cell proliferation, migration, and invasion through the miR-106a-5p/DDX5/AKT axis in CRC while inhibiting the malignant progression by regulating the expression levels of miR-183-5p and TUSC2 in glioma. Importantly, recent studies have demonstrated that circ-EGFR deprivation suppresses tumorigenicity in brain tumor cells and enhances the efficacy of nimotuzumab in glioblastoma. Based on these exciting recent studies, circ-EGFR might be a promising predictor for guiding anti-EGFR treatment in mCRC patients, offering hopeful insights for personalized therapy.

In our present study, we discovered that tissue-based circ-EGFR serves as a promising predictor for guiding anti-EGFR treatment in patients with mCRC. Specifically, through systematic and comprehensive cell culture and animal studies, we demonstrated that circ-EGFR functions as a sponge for miR-942-3p, which in turn regulates the expression of GAS1 and the Hedgehog (Hh) signaling pathway, thereby promoting sensitivity to cetuximab treatment in CRC. Notably, we successfully translated this tissue-based biomarker into a liquid biopsy tool for predicting responses to anti-EGFR therapy. Collectively, our findings offer significant promise for improving patient selection and management of this fatal malignancy.

## Results

### Circ-EGFR expression is elevated in cetuximab-sensitive colorectal cancer cells

Given the pivotal role of EGFR in cancer development and its close association with the efficacy of EGFR tyrosine kinase inhibitors, we investigated the influence of EGFR-derived circRNAs on cetuximab treatment. First, we evaluated cetuximab sensitivity in eight *KRAS* WT CRC cell lines (C2BBE1, CAR-1, COLO-320-HSR, SW48, DiFi, NCI-H716, SNU-C1, and HT29) from the Genomics of Drug Sensitivity in Cancer (GDSC) database (Table EV1, https://www.cancerrxgene.org/). Based on (IC$_{50}$) values, we identified two cetuximab-sensitive cell lines (IC$_{50}$ of DiFi: 1.26 μM and SW48: 55.54 μM) and two cetuximab-resistant cell lines (IC$_{50}$ of SNU-C1: 2589.34 μM and HT29: 647.66 μM). Additionally, the CaCO2 cell line (*KRAS* WT) was also reported as cetuximab-resistant (IC$_{50}$ > 1000 μM) (Georgiou et al, 2021; Song et al, 2014); hence, these cell lines were selected for subsequent experiments. Regardless, to definitively confirm the sensitivity of these five cell lines to cetuximab (DiFi, SW48, CaCO2, SNU-C1, and HT29), we conducted cell viability assays which confirmed the growth-inhibitory effects of cetuximab (IC$_{50}$ of DiFi: <5 μg/mL, SW48: 34.77 μg/mL, CaCO2: N/A, SNU-C1: N/A, and HT29: N/A; Fig. EV1). These results indicated that DiFi and SW48 cells were indeed sensitive to cetuximab treatment, whereas HT29, SNU-C1, and CaCO2 cells exhibited significant resistance to this anti-EGFR drug. Subsequently, we examined the expression of thirteen circ-EGFR transcripts by RT-qPCR assays in these cetuximab-sensitive and resistant cell lines. Among these, hsa_circ_0080230 showed the most significant upregulation in cetuximab-sensitive cells compared to resistant ones after cetuximab treatment (Fig. 1A). According to the University of California, Santa Cruz (UCSC) database (https://genom.ucsc.edu/), circ-EGFR (hsa_circ_0080230)

is mapped to chr7: 55231425-55249171 and derived from exons 14 to 20 of the protein-coding gene EGFR, and its circular structure was confirmed by Sanger sequencing (Fig. 1B).

Next, we characterized the expression pattern of circ-EGFR in CRC cell lines. For these PCR assays, unique primers were designed, which required divergent and convergent primers for the specific amplification of circ-EGFR from the complementary DNA (cDNA) and genomic DNA (gDNA). Considering that DiFi and SW48 expressed relatively high levels of circ-EGFR, these cell lines were selected for the following experiments. Agarose gel electrophoresis revealed that the circ-EGFR signal was only detectable using the divergent primers in cDNA instead of gDNA, indicating the specificity of our primer design and the fact that circ-EGFR exhibits a reverse loop structure (Fig. 1C). Moreover, compared with the linear host transcripts of EGFR, circ-EGFR exhibited increased resistance to RNase-R exonuclease digestion, confirming the circular nature of circ-EGFR that is devoid of a 5'-cap and a 3'-tail in DiFi and SW48 cells (Fig. 1D). Furthermore, we explored the cellular localization of circ-EGFR in CRC by RNA fluorescence in situ hybridization (FISH) analysis. The results revealed that circ-EGFR was primarily located in the cytoplasm (Fig. 1E). Taken together, our findings demonstrated that circ-EGFR is an endogenous circular RNA in CRC cells and exhibits significantly elevated expression in cetuximab-sensitive cells.

### Expression levels of circ-EGFR serve as a promising biomarker for risk stratification and response prediction to cetuximab in mCRC

Given the lack of prior studies on circ-EGFR expression and cetuximab response in metastatic colorectal cancer (mCRC), we first undertook these experiments in a clinical trial cohort of patients with mCRC. To achieve this, we enrolled 45 patients with mCRC, and tumor specimens from these patients were collected prior to the initiation of cetuximab therapy. Thereafter, we divided these cases into two groups: the cetuximab response group (n = 25), comprising cases that did not experience tumor progression up to 9 months of treatment, and the cetuximab non-response group (n = 20), in which tumor progression was observed within 9 months of treatment. Interestingly, we observed a significant upregulation of circ-EGFR expression in the cetuximab response group vis-à-vis the non-response group (Fig. 1F). Receiver operating characteristic (ROC) analysis revealed the robust predictive ability of circ-EGFR, as evidenced by a remarkable area under the curve (AUC) value of 76.8% [95% confidence interval (CI): 61.2–92.3%] (Fig. 1G). The waterfall plot illustrates the ability of circ-EGFR to identify responders to cetuximab treatment, with 22 out of 25 cases (88%) correctly identified as true positives and 13 out of 20 cases (65%) as true negatives in this clinical trial cohort (Fig. EV2).

Our research aimed to evaluate the clinical impact and application of the circ-EGFR molecular marker. We found that among the 45 mCRC patients who received cetuximab treatment, 20 showed no response, resulting in an overtreatment ratio of 44.4% as per the current treatment guidelines. However, when we stratified the patients based on the circ-EGFR expression, we observed promising results. Among those predicted to respond to cetuximab treatment, only 8 cases did not show a measurable response, leading to a reduced overtreatment ratio of 17.8% based on the circ-EGFR-based approach. This significant reduction in

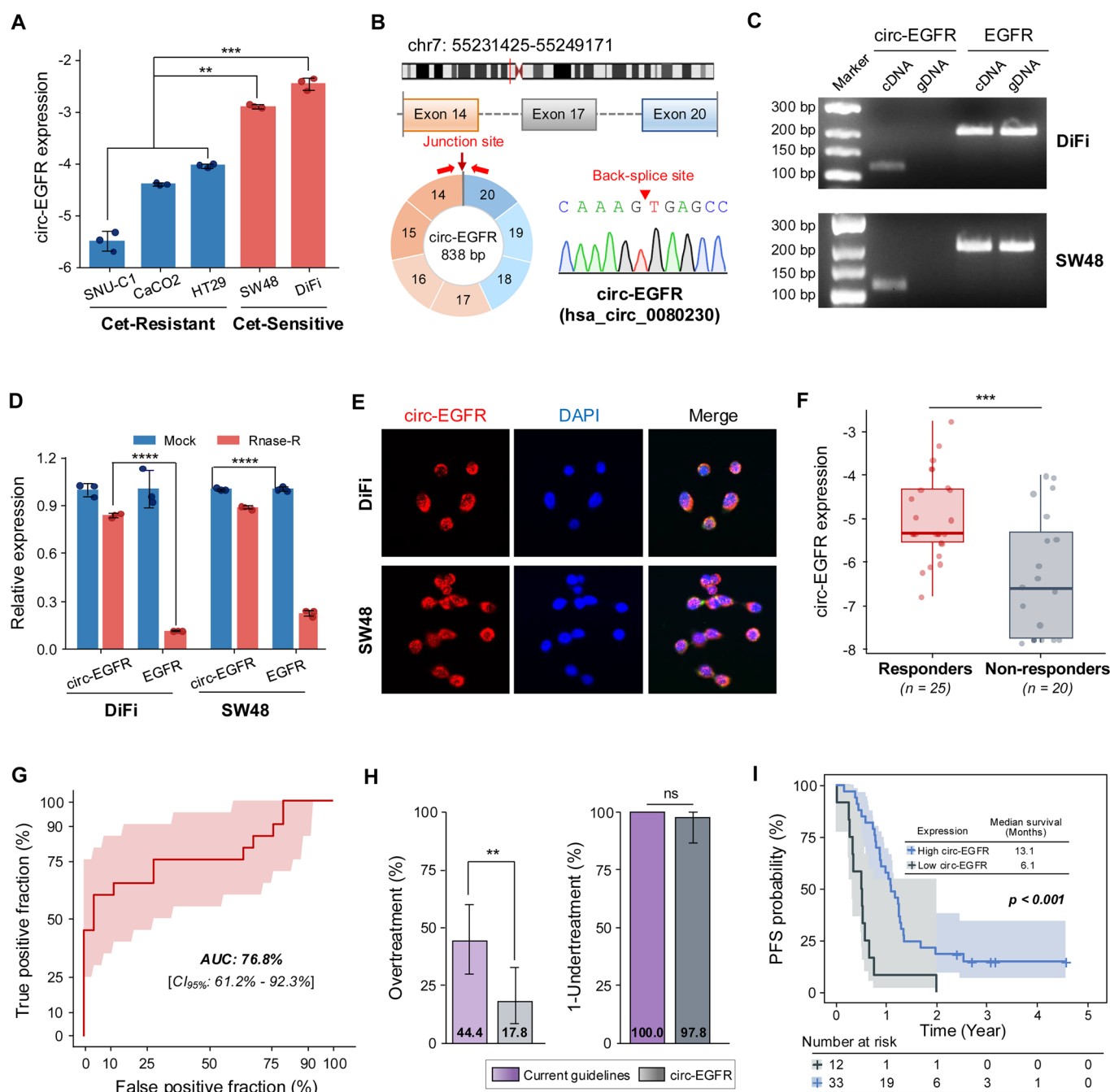

unnecessary treatment (Fig. 1H, left panel) is a promising finding for the future management of patients with metastatic colorectal cancer (mCRC). In contrast, our circ-EGFR approach predicted only 1 out of 45 cases as a false non-responder, with an undertreatment ratio of 2.2% (1/45). There were non-significant (ns) differences in the undertreatment ratios between the current treatment guidelines and the circ-EGFR-based approach (Fig. 1H, right panel). Furthermore, Kaplan–Meier survival analysis demonstrated that the PFS following cetuximab treatment in the circ-EGFR-low group was significantly shorter compared to the circ-EGFR-high group, with median survival times of 6.1 and 13.1 months, respectively (Fig. 1I, $p < 0.001$). In conclusion, our

data underscores the role of circ-EGFR in risk stratification and response prediction in mCRC patients treated with cetuximab, offering hope for more effective and personalized cancer treatments in the future.

## Circ-EGFR enhances the sensitivity of CRC cells to cetuximab

To elucidate the biological function of circ-EGFR in cetuximab response, we selected cetuximab-resistant CRC cell lines, CaCO2 and SNU-C1, for the subsequent studies. We established stable cell lines by overexpressing (OE) circ-EGFR in CaCO2 and SNU-C1

Figure 1. Circ-EGFR serves as a promising predictive biomarker for cetuximab response in CRC.

(A) Relative circ-EGFR expression in cetuximab-resistant (CaCO2, SNU-C1, and HT29) and cetuximab-sensitive (SW48 and DiFi) cell lines. Cetuximab-resistant (Cet-R) vs. SW48: $p = 0.001$, Cet-R vs. DiFi: $p = 0.0003$. (B) Schematic diagram of the formation of circ-EGFR. Circ-EGFR was back-spliced from exons 14 to 20 of EGFR, and its back-splice sequence was confirmed by Sanger sequencing. (C) Agarose gel electrophoresis with divergent or convergent primers indicated that circ-EGFR existed in cDNA but not in gDNA in DiFi and SW48 cell lines. (D) Relative abundance of circ-EGFR and linear-EGFR in DiFi and SW48 cells treated with RNase-R, as determined by RT-qPCR. circ-EGFR vs. EGFR in two cells: $p < 0.0001$. (E) FISH assay confirmed the localization of circ-EGFR in DiFi and SW48 cells. (F) The relative expression level of circ-EGFR was evaluated by RT-qPCR in a cohort of 45 cases with varying responses to cetuximab treatment. Boxplot (P25 − 1.5×IQR; P25; median; P75; P75 + 1.5×IQR, Tukey method) shows the distribution of circ-EGFR expression levels in tissue samples from responders and non-responders. Responders: minimum = −6.772, maximum = −2.755, median = −5.325, whisker low = −6.772, whisker high = −2.755; non-responders: minimum = −7.841, maximum = −4.001, median = −6.601, whisker low = −7.841, whisker high = −4.001; Responders vs. non-responders: $p = 0.0008$. (G) ROC curve analysis evaluates the performance of the circ-EGFR biomarker in the clinical trial cohort. ROC curves are shown as 95% CIs. (H) The ratios of overtreatment and 1-undertreatment for the current guideline and circ-EGFR-based approach were demonstrated in the clinical trial cohort. Overtreatment: $p = 0.007$, 1-undertreatment: $p = 0.317$. (I) The Kaplan–Meier curve exhibited correlations between circ-EGFR expression and PFS of CRC patients in the clinical cohort. High vs. low expression: $p = 0.0002$. Data were representative of at least three independent biological replicates. Bars represent the mean and the error bars indicate SD. Statistical analyses of differences between groups were performed using two-tailed Student's t-tests. Kaplan–Meier survival analyses were evaluated using the log-rank test. $*p < 0.05$, $**p < 0.01$, $***p < 0.001$, $****$ $p < 0.0001$, ns: not significant. Source data are available online for this figure.

cells, confirmed by RT-qPCR assays (Fig. 2A,B, left panel). MTT assay revealed that the circ-EGFR overexpression significantly inhibited cell proliferation in both CaCO2 and SNU-C1 cells treated with cetuximab (cetuximab concentration for CaCO2 and SNU-C1: 150 μg/mL) (Fig. 2A,B, right panel). The colony formation assay also demonstrated that circ-EGFR overexpression significantly inhibited colony formation potential in CRC cells following cetuximab treatment (Fig. 2C), indicating that circ-EGFR mediates the response to cetuximab in CRC cells. Next, we investigated the impact of circ-EGFR on cellular apoptosis in CRC cells after cetuximab treatment. Flow cytometry assay revealed that the overexpression of circ-EGFR substantially increased the proportion of apoptotic cells in both CaCO2 and SNU-C1 cell lines following cetuximab treatment (Fig. 2D).

Given the relatively elevated expression levels of circ-EGFR in the SW48 and DiFi cell lines, we further examined whether inhibition of circ-EGFR would induce resistance to cetuximab in these cell lines. Subsequently, we constructed stable cell lines with circ-EGFR silencing in DiFi and SW48 cells using two circ-EGFR-specific lentivirus-based short hairpin RNAs (shRNAs). The inhibition efficacy of circ-EGFR was successfully validated via RT-qPCR assays (Fig. EV3A, left panel). The cell viability and colony formation assay showed that depletion of circ-EGFR significantly increased cell proliferation and colony formation upon treatment with cetuximab (Fig. EV3A,B). We then investigated the impact of circ-EGFR on the migratory and invasive capabilities of CRC cells after treatment with cetuximab. Wound healing assays demonstrated that circ-EGFR depletion promoted migration of DiFi and SW48 cells treated with cetuximab (Fig. EV3C). As a further verification, we performed an invasion assay with Matrigel and observed that circ-EGFR knockdown significantly increased the invasive abilities in CRC cells under cetuximab treatment (Fig. EV3D). Taken together, these data demonstrate that circ-EGFR enhances cetuximab sensitivity in CRC cells, suggesting that circ-EGFR may be a potential predictor of cetuximab response in metastatic colorectal cancer (mCRC).

## Overexpression of circ-EGFR promotes response to cetuximab in an animal model

To further confirm the effect of circ-EGFR on response to cetuximab in vivo, we established a subcutaneous xenograft mouse model using

the cetuximab-resistant HT29 cell line (*KRAS* WT). As illustrated in the study design schema (Fig. EV4A), we subcutaneously inoculated $5 \times 10^6$ HT29 cells stably overexpressing circ-EGFR ($n = 10$) or mock controls ($n = 10$) into nude mice. Once the xenograft tumors became visible (~1 week later), we treated nude mice carrying xenograft tumors with either saline or cetuximab (40 mg/kg per mouse, administered biweekly) for 2 weeks. In brief, the nude mice were divided into four groups of five mice each: mice inoculated with HT29 cell line transfected with vector control and treated with cetuximab (Ctrl Cet), mice inoculated with HT29 cell line transfected with vector control and treated with saline (Ctrl Saline), mice inoculated with circ-EGFR overexpressing HT29 cells and treated with cetuximab (OE Cet), and mice inoculated with circ-EGFR overexpressing HT29 cells and treated with saline (OE Saline). Finally, the animals were sacrificed ~21 days following the subcutaneous injection. Upon treatment with cetuximab, it was observed that the antitumor efficacy was decreased significantly in the circ-EGFR-OE group compared with the vector control group (Fig. 2E,F). No significant body weight loss was observed in any group (Fig. EV4B). Our results once again indicated that overexpression of circ-EGFR promoted the therapeutic efficacy of cetuximab in CRC cells (Fig. 2G).

Given that cetuximab is primarily used in patients with advanced CRC, we established a liver metastasis model to better recapitulate the role of circ-EGFR in mediating therapeutic response. HT29 cells with mock controls or circ-EGFR overexpression were injected into the spleens of mice to facilitate efficient hepatic colonization. Liver metastases were monitored by in vivo bioluminescence imaging on day 7 post-injection. Upon confirmation of liver metastasis formation, mice were treated with either saline or cetuximab (40 mg/kg per mouse, administered biweekly) for 2 weeks. At the end of the treatment period, bioluminescence imaging was performed to evaluate tumor burden. Compared to the cetuximab-treated control group, mice bearing circ-EGFR-overexpressing tumors showed a significant reduction in liver bioluminescence intensity (Fig. 3A, upper panel). On day 30 post-injection, mice were euthanized, and liver tissues were harvested for histological and tumor burden analysis. Notably, circ-EGFR overexpressing mice exhibited a significantly lower liver tumor weight compared to the control group following cetuximab treatment (Fig. 3A, lower panel). Histopathological analysis by H&E staining revealed substantial architectural disruption and numerous metastatic lesions, whereas those retained more intact architecture and displayed a notable reduction in metastatic burden following anti-EGFR treatment (Fig. 3B).

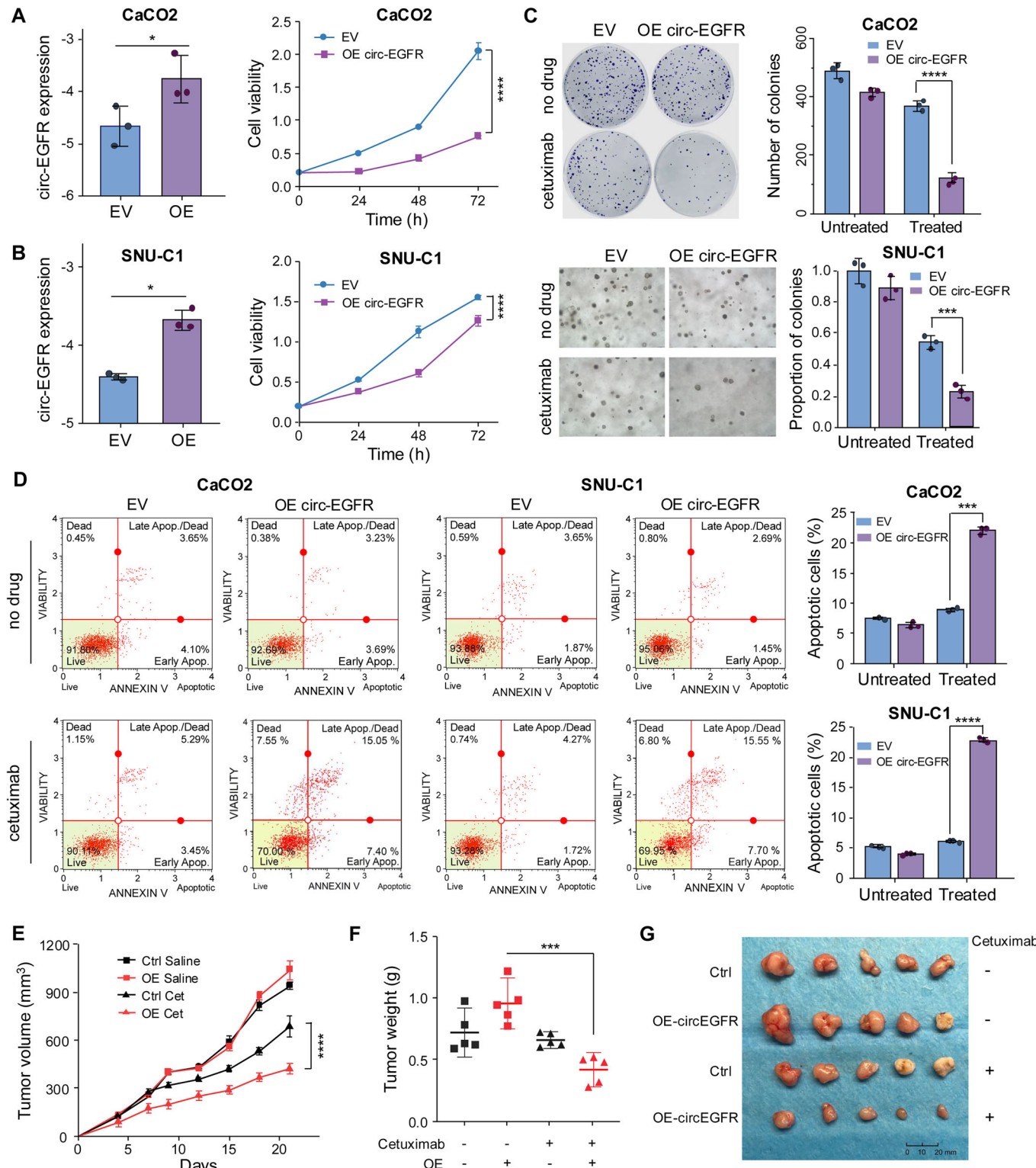

## Circ-EGFR acts as a miRNA sponge for miR-942-3p in colorectal cancer

Given that one of the primary gene regulatory functions of circRNAs is to impact tumor pathogenesis by serving as competing endogenous RNAs (ceRNAs) through their interaction with miRNAs (Han et al, 2017; Kleaveland et al, 2018; Yang et al, 2018), we explored the ability of circ-EGFR to act as a molecular sponge for miRNAs in CRC. Firstly, analysis of publicly available datasets indicated a significant level of AGO2, an essential mediator of circRNA-miRNA interaction (Hansen

**Figure 2.   Circ-EGFR overexpression promotes the response of CRC cells to cetuximab in vitro and in vivo.**

(A, B) The relative expression level of circ-EGFR in CaCO2 (A) and SNU-C1 (B) CRC cell lines transfected with circ-EGFR overexpression (OE) plasmid or empty vector (EV) (left panel). OE vs. EV in CaCO2: $p = 0.020$, SNU-C1: $p = 0.012$ (Two-tailed Student's $t$-tests). Assessment of cell proliferation capacity by MTT assay in circ-EGFR overexpression or EV group from CaCO2 (A) and SNU-C1 (B) cell lines after 48 h treatment with cetuximab (right panel). CaCO2: $p < 0.0001$, SNU-C1: $p < 0.0001$ (two-way ANOVA test). (C) Colony formation assay showed the ability of colony formation in CaCO2 and SNU-C1 cells transfected with circ-EV/circ-EGFR-OE plasmids following treatment with cetuximab for 48 h. Two-tailed Student's $t$-tests were used to compare OE and EV groups in cetuximab-treated CaCO2 ($p < 0.0001$) and SNU-C1 ($p = 0.0008$) cells. (D) Representative images of cells undergoing apoptosis that stained for the annexin V assay in CaCO2 and SNU-C1 cells transfected with circ-EV/circ-EGFR-OE plasmids with or without cetuximab treatment. OE vs. EV in treated CaCO2: $p = 0.0002$, SNU-C1: $p < 0.0001$ (two-tailed Student's $t$-tests). (E, F) Tumor volume (E) and tumor weight (F) of nude mice injected with circ-EGFR and mock plasmids in each treatment group were measured at different time points after inoculation. Tumor volume: $p < 0.0001$ (two-way ANOVA), tumor weight $p = 0.0003$ (student's $t$-test). (G) Images of subcutaneous xenograft tumors derived from HT29 cells transfected with circ-EGFR and mock plasmids in each treatment group. Data were representative of at least three independent biological replicates. Bars represent the mean, and error bars indicate SD. Ctrl Cet: mice inoculated with the HT29 cell line transfected with vector control and treated with cetuximab, Ctrl Saline: mice inoculated with the HT29 cell line transfected with vector control and treated with saline, OE Cet: mice inoculated with circ-EGFR overexpressing HT29 cells and treated with cetuximab, OE Saline: mice inoculated with circ-EGFR overexpressing HT29 cells and treated with saline. *$p < 0.05$, **$p < 0.01$, ***$p < 0.001$, **** $p < 0.0001$, ns not significant. Source data are available online for this figure.

et al, 2013), binding within the circ-EGFR region. RIP-qPCR assay subsequently validated that circ-EGFR was significantly enriched in the AGO2 group instead of IgG control in CRC, indicating that circ-EGFR likely binds to RNA-induced silencing complex (RISC) and sponge corresponding miRNAs (Fig. 4A). In addition, using circBank, circInteractome, and circAtlas databases, we successfully identified a panel of five miRNAs (hsa-miR-942-3p, hsa-miR-665, hsa-miR-609, hsa-miR-296-5p, and hsa-miR-1184) that overlapped between these datasets (Fig. 4B). Next, we evaluated the expression levels of these 5 miRNAs following circ-EGFR overexpression in CaCO2 and SNU-C1 cells using RT-qPCR assays. In these experiments, we observed that miR-942-3p emerged as the most consistent miRNA, exhibiting significantly lower expression following the overexpression of circ-EGFR in both cell lines (Fig. 4C).

Next, we determined whether circ-EGFR could directly bind miR-942-3p by pull-down assay using a specific biotin-labeled circ-EGFR probe. RNA extracted from the pull-down assay was subjected to gel electrophoresis and RT-qPCR analysis, revealing that the circ-EGFR probe was robustly enriched for circ-EGFR compared to the oligo probe in both CaCO2 and SNU-C1 cells (Fig. 4D,E). The RNA pull-down assay demonstrated that miR-942-3p was significantly captured by a biotin-labeled circ-EGFR probe in CaCO2 and SNU-C1 cells (Fig. 4F). To further elucidate the potential binding site between circ-EGFR and miR-942-3p in CRC, we constructed WT and MUT dual luciferase reporter plasmids of circ-EGFR (Fig. 4G). Luciferase assays revealed that miR-942-3p mimics could significantly suppress circ-EGFR luciferase activity (Fig. 4H), whereas no significant change was observed in the MUT construct, indicating that miR-942-3p is a putative target of circ-EGFR. Finally, we examined circ-EGFR expression in cetuximab responders and non-responders within mCRC patient cohorts from the GEMCAD trials using the FISH assay. The results indicated that circ-EGFR colocalized with miR-942-3p in mCRC patients, and more notably, was significantly upregulated in the cetuximab response group (Fig. 4I). Collectively, these results strongly support the idea that circ-EGFR directly binds to miR-942-3p and potentially serves as its sponge in exerting its function in CRC.

Next, we aimed to validate the regulatory impact of circ-EGFR on miR-942-3p, which promotes cetuximab response. Through RT-qPCR analysis, we successfully co-transfected SNU-C1 and CaCO2 cells with circ-EGFR and miR-942-3p overexpression plasmids (Fig. 5A,B, left panel). Next, we performed cell proliferation assays to examine the effects of circ-EGFR and miR-942-3p on CRC cells. MTT assay

revealed that miR-942-3p overexpression significantly abolished the inhibitory effect of circ-EGFR on cell growth under cetuximab treatment in SNU-C1 and CaCO2 cells (Fig. 5A,B, right panel). Colony formation assays confirmed that the response of circ-EGFR to cetuximab was abrogated by the introduction of miR-942-3p in SNU-C1 and CaCO2 cells (Fig. 5C). Additionally, we observed that miR-942-3p could effectively abolish the apoptosis ability of circ-EGFR-OE in CRC cells (Fig. EV5). Following cetuximab treatment, the overexpression of circ-EGFR resulted in the upregulation of Bax, cleaved caspase 3, and cleaved caspase 9 expression in CaCO2 and SNU-C1 cells, as indicated by the protein levels of apoptosis-related genes. However, these effects were reversed by miR-942-3p over-expression (Fig. 5D). Collectively, these results further support our hypothesis that circ-EGFR has a significant impact on cetuximab sensitivity by regulating its downstream target, miR-942-3p.

## GAS1 is a direct target of miR-942-3p and is regulated by circ-EGFR in CRC

To identify downstream gene targets responsible for mediating the function of miR-942-3p, we conducted an extensive analysis to identify such targets based on miR-942-3p binding, as determined by the TargetScan, miRTarBase, miRDB, and mirDIP search tools. As illustrated in Fig. 6A, we identified three gene targets (GAS1, RNF44, and PGGT1B) exhibiting significant potential to bind to miR-942-3p. Next, RT-qPCR analysis revealed that overexpression of circ-EGFR significantly increased the mRNA expression of GAS1 but not RNF44 and PGGT1B in CRC cell lines (Fig. 6B). Furthermore, circ-EGFR overexpression also led to increased GAS1 protein levels in CaCO2 and SNU-C1 cells, while miR-942-3p downregulated GAS1 in SW48 and DiFi cells (Fig. 6C). In summary, these results revealed that circ-EGFR is a miR-942-3p sponge to regulate GAS1 expression in CRC.

To further demonstrate whether GAS1 is a direct target of miR-942-3p, we performed luciferase reporter assays to investigate the interactions between miR-942-3p and the 3' UTR of GAS1. As shown in Fig. 6D, the full-length 3′UTR of GAS1 with WT or MUT miR-942-3p binding sites was subcloned into PGL3-CMV-LUC vectors. Luciferase assay revealed that miR-942-3p significantly decreased the relative luciferase activity in the WT group but not the MUT group (Fig. 6E).

Accumulating evidence suggests that GAS1 exhibits a high affinity for Hh, indicating its potential role in regulating this important growth-regulatory cell signaling pathway (Lee et al, 2001; Wierbowski et al,

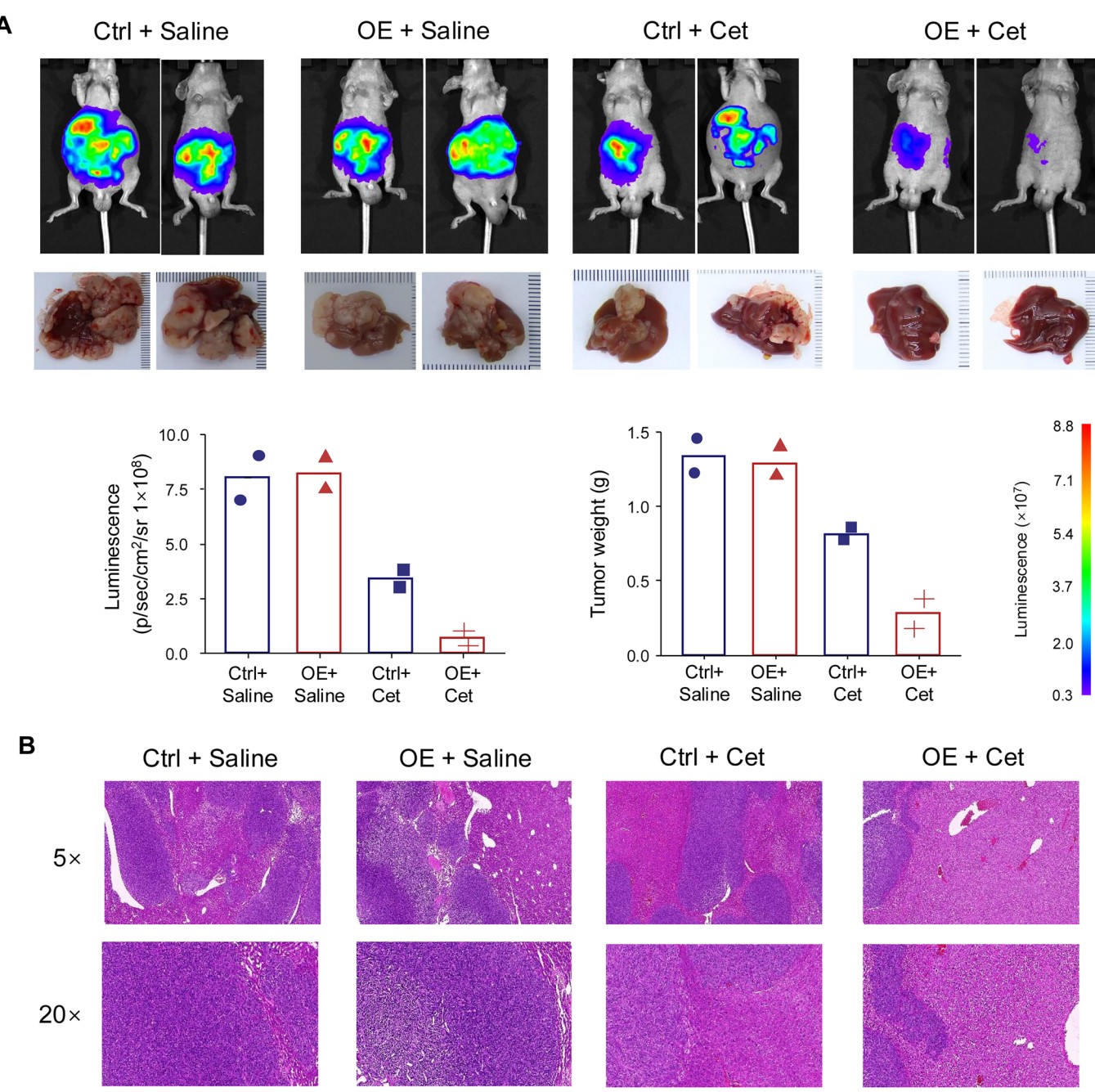

**Figure 3.   Circ-EGFR improves cetuximab efficacy in a hepatic metastasis mouse model.**

(A) Bioluminescence imaging of 8 mice with liver metastases 21 days after splenic injection of HT29 cells (mock control and circ-EGFR overexpression, upper panel), and corresponding metastatic livers were harvested for further analysis (lower panel). (B) Representative H&E-stained liver sections from mice in different experimental groups, shown at 5× and 20× magnification. Ctrl + Saline: mock control HT29 cells treated with saline, OE + Saline: circ-EGFR overexpressing HT29 treated with saline, Ctrl + Cet: mock control HT29 treated with cetuximab, OE + Cet: circ-EGFR overexpressing HT29 treated with cetuximab. Source data are available online for this figure.

2020). The Hh signaling pathway is pivotal in tumor growth and development (Rubin and de Sauvage, 2006). We next sought to evaluate the regulatory effect of the circ-EGFR/miR-942-3p/GAS1 axis on the Hh signaling pathway by characterizing the expression levels of key pathway regulators, including SHH, SMO, and GLI1, by western immunoblotting. As expected, it was observed that circ-EGFR overexpression upregulated the protein expression of GAS1, SHH,

SMO, and GLI1 in the CaCO2 and SNU-C1 cell lines (Fig. 6F). More importantly, these effects could be abolished by miR-942-3p over-expression (Fig. 6F). To further validate the involvement of the Hh pathway in modulating cetuximab response in CRC, we conducted in vitro assays using two FDA-approved inhibitors, Sonidegib (SON), and Vismodegib (VIS), to evaluate their impact on cetuximab sensitivity. Both inhibitors reduced cell growth in a dose-dependent

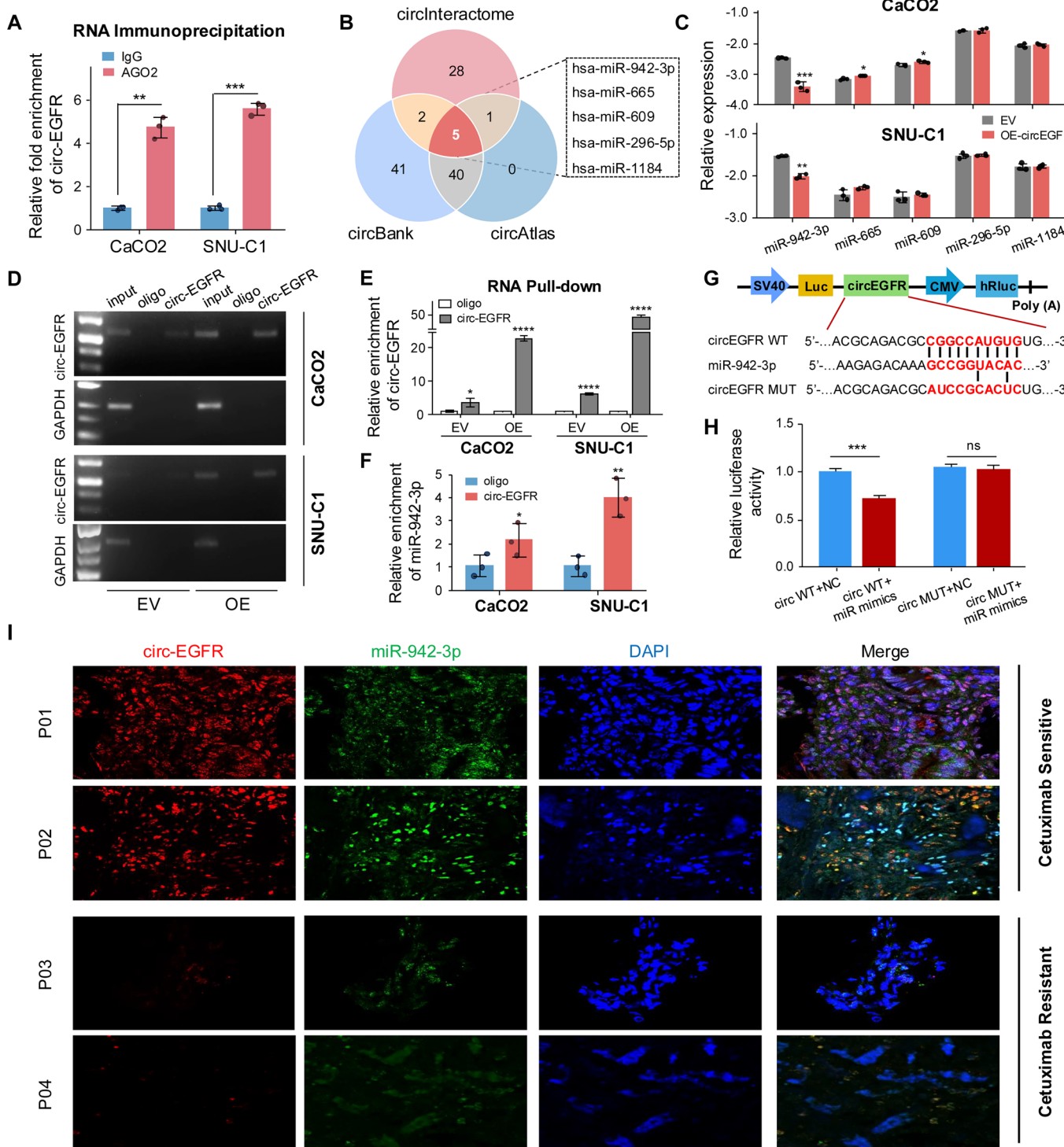

manner in CaCO2 and SNU-C1 cells, with IC$_{50}$ values of 63.31 nM (SON) and 102.10 nM (VIS) in CaCO$_2$, and 42.22 nM (SON), and 103.90 nM (VIS) in SNU-C1 (Fig. EV6A). Next, we evaluated the combined effects of cetuximab and Hh inhibitors using the SynergyFinder. The combination of Cetuximab and Sonidegib achieved inhibition rates exceeding 80%, while Cetuximab and Vismodegib resulted in inhibition rates over 75% (Fig. EV6B). Both combinations showed ZIP synergy scores greater than 10 (Fig. EV6C), indicating

strong synergistic interactions in CRC. In addition, the inhibition rates of the two CRC cell lines treated with various combinations of Cetuximab and Hh inhibitors were analyzed via Compusyn software to calculate the Combination Index values. The results showed that most of the data points were positioned below the line of additive effects (Combination index = 1), suggesting a potentiating effect of Cetuximab and Hh inhibitor (Fig. EV6D,E). Finally, western blot analysis revealed that while cetuximab or Hh inhibitor alone partially suppressed the

◄ **Figure 4. Circ-EGFR acts as a miRNA sponge of miR-942-3p.**

(A) RNA immunoprecipitation assay was performed with cell lysate of CaCO2 and SNU-C1 cells using either anti-AGO2 or IgG as the immunoprecipitating antibody. CaCO2 $p = 0.005$, SNU-C1 $p = 0.0005$ (student's $t$-test). (B) Venn diagram showing the candidate miRNAs predicted to be the binding targets of circ-EGFR by circBank, circInteractome, and circAtlas. (C) Relative expression of 5 candidate miRNAs after overexpression of circ-EGFR in CaCO2 and SNU-C1 cell lines. CaCO2: miR-942-3p $p = 0.0005$, miR-665 $p = 0.015$, miR-609 $p = 0.034$, miR-296-5p $p = 0.714$, miR-1184 $p = 0.633$. SNU-C1: miR-942-3p $p = 0.003$, miR-665 $p = 0.124$, miR-609 $p = 0.521$, miR-296-5p $p = 0.650$, miR-1184 $p = 0.894$ (student's $t$-test). (D, E) Gel electrophoresis (D) and RT-qPCR analysis (E) confirmed the efficiency of the RNA pull-down assay by comparing the relative enrichment of circ-EGFR between the circ-EGFR probe and oligo probe. circ-EGFR vs. oligo in CaCO2 EV: $p = 0.023$, CaCO2 OE: $p < 0.0001$, SNU-C1 EV: $p < 0.0001$, SNU-C1 OE: $p < 0.0001$ (student's $t$-test). (F) The relative enrichment of miR-942-3p in cell lysates pulled down by oligo or circ-EGFR probes was examined by RT-qPCR in CaCO2 and SNU-C1 cells. CaCO2 $p = 0.014$, SNU-C1 $p = 0.006$ (student's $t$-test). (G) Schematic illustration of circ-EGFR WT and mutant (MUT) luciferase reporter vectors were shown. (H) The luciferase activity of circ-EGFR WT or circ-EGFR MUT after transfection with miR-942-3p mimics in 293T cells. WT $p = 0.0001$, MUT $p = 0.357$ (student's $t$-test). (I) Representative fluorescence images of circ-EGFR and miR-942-3p in tissues derived from metastatic colorectal cancer (mCRC) patients with varying responses to cetuximab were obtained using FISH. Data were representative of at least three independent biological replicates. Bars represent the mean, and error bars indicate SD. *$p < 0.05$, **$p < 0.01$, ***$p < 0.001$, **** $p < 0.0001$, ns not significant. Source data are available online for this figure.

expression of key Hh pathway components (SHH, SMO, and GLI1), the combination treatment resulted in a more pronounced downregulation of these proteins (Fig. EV6F). These results demonstrate that circ-EGFR functions as a sponge for miR-942-3p, affecting cetuximab response by regulating the GAS1/Hh pathway. These findings highlight the potential of combining cetuximab with hedgehog (Hh) inhibitors as a therapeutic strategy to enhance treatment efficacy in metastatic colorectal cancer (mCRC).

### Circ-EGFR influences cetuximab response via the miR-942-3p/GAS1 axis in CRC

Building upon our previous findings, we investigated the correlation between circ-EGFR, miR-942-3p, and GAS1 in the clinical cohort of 45 mCRC patients treated with cetuximab. In support of our findings in the cell line and murine model, miR-942-3p exhibited significant upregulation (Fig. 7A), while GAS1 demonstrated downregulation in non-responder patients (Fig. 7B). Moreover, we observed a significant positive correlation between circ-EGFR and GAS1 ($r_s = 0.34$, $p = 0.036$, Fig. 7C) in this clinical trial-derived cohort of mCRC patients. While not statistically significant, circ-EGFR and miR-942-3p ($r_s = -0.24$, $p = 0.138$, Fig. 7D), as well as GAS1 and miR-942 ($r_s = -0.23$, $p = 0.161$, Fig. EV7), exhibited a trend toward a negative correlation. Furthermore, Kaplan–Meier analysis revealed that high expression of miR-942-3p exhibited a significantly poorer PFS after treatment with cetuximab (Fig. 7E, 9-month PFS rates for high vs. low: 43.5 vs. 77.3%, $p = 0.032$). Conversely, CRC patients with higher GAS1 expression showed a trend toward improved overall prognosis (Fig. 7F, 9-month PFS rates for high vs. low: 73.9 vs. 50.0%, $p = 0.160$). Collectively, our findings reveal significant correlations between circ-EGFR, miR-942-3p, and GAS1 in mCRC treated with cetuximab, underscoring their potential as predictive biomarkers for treatment response in clinical practice.

### Translation of tissue-based circ-EGFR expression into a liquid biopsy assay for predicting response to cetuximab in patients with mCRC

Liquid biopsies provide a minimally invasive approach for characterizing the progression of disease, specifically mCRC, and inform patient treatment decisions. Such assays help avoid the overtreatment of patients with therapies that offer no clinical benefit and can often be associated with treatment-related toxicities. Accordingly, we investigated the feasibility of translating our tissue-based circ-EGFR into a liquid biopsy assay in an independent cohort from a clinical trial. Consistent with the tissue-based clinical trial cohort, the threshold for disease progression was set at 9 months ($n = 112$, 79 responders and 33 non-responders). The expression levels of circ-EGFR were readily detectable in plasma, and we observed a significant upregulation of circ-EGFR expression in responders to anti-EGFR therapy (Fig. 8A). ROC analysis suggested the robust performance of circ-EGFR in predicting anti-EGFR therapy response in mCRC (Fig. 8B, AUC: 76.9%, 95% CI: 66.2–87.6%). Patients were stratified into circ-EGFR-high and -low groups based on Youden's index cutoff ($-1.151$). Moreover, Kaplan–Meier survival analysis revealed a significantly shorter median PFS for patients with circ-EGFR-low expression compared to those with circ-EGFR-high expression (8.0 vs. 14.0 months) and a lower 9-month PFS rate (42.7 vs. 86.1%) following anti-EGFR therapy treatment (Fig. 8C).

Next, we aimed to determine whether circ-EGFR could serve as a reliable predictor of cetuximab response by evaluating the correlation between circ-EGFR and maximum tumor shrinkage (MTS), a key indicator of treatment response. We observed a significantly positive relationship between circ-EGFR expression and MTS (Fig. 8D). Finally, we assessed the potential clinical significance of the circ-EGFR predictive biomarker. The circ-EGFR-based approach markedly reduced the overtreatment ratio compared to the current guideline approach (Fig. 8E, left panel). In contrast, these two approaches have no significant difference in undertreatment rates (Fig. 8E, right panel). Taken together, these findings underscore the clinical utility of circ-EGFR-based liquid biopsy assay for non-invasive prediction of response to anti-EGFR agents in mCRC.

## Discussion

Cetuximab is the most effective anti-EGFR drug for the treatment of patients with mCRC. When bound to the extracellular domain of the EGFR, cetuximab blocks endogenous ligand binding, thereby inhibiting tumor cell proliferation and promoting cancer cell apoptosis (Sobrero et al, 2008). Hence, considering the promising efficacy of cetuximab as an anticancer therapy, this regimen has been increasingly utilized as a first-line treatment for CRC, particularly in cases with advanced CRC (Morris et al, 2023). Despite the notable clinical efficacy of cetuximab, including its ability to enhance the overall PFS and overall survival (OS), improve the quality of life of patients, and exhibit low side effects,

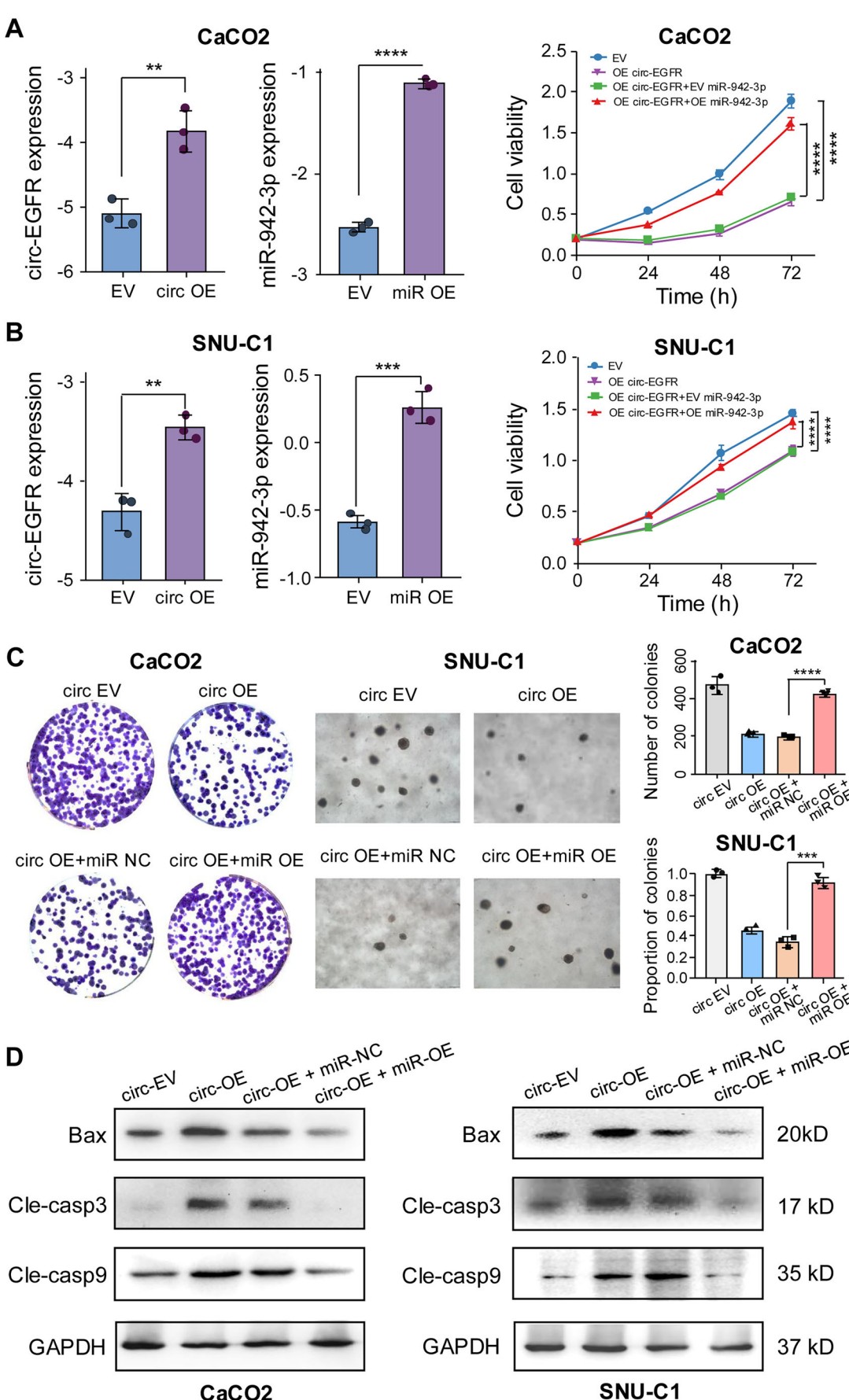

**Figure 5.  Overexpression of circ-EGFR could enhance cetuximab sensitivity by regulating miR-942-3p.**

(A, B) The relative expression level of circ-EGFR and miR-942-3p in CaCO2 (A) and SNU-C1 (B) CRC cell lines transfected with circ-EGFR, miR-942-3p, or EV (left panel). circ-EGFR expression in CaCO2: $p = 0.005$, miR-942-3p in CaCO2: $p < 0.0001$, circ-EGFR in SNU-C1: $p = 0.003$, miR-942-3p in SNU-C1: $p = 0.0003$ (student's $t$-test). Cell proliferation capacity by MTT assay in CaCO2 (A) and SNU-C1 (B) cell lines transfected with circ-EGFR and/or miR-942-3p plasmids (right panel). CaCO2 $p < 0.0001$, SNU-C1 $p < 0.0001$ (two-way ANOVA). (C) Colony formation assay showed the ability of colony formation in CaCO2 and SNU-C1 cells co-transfected with circ-EV/circ-EGFR-overexpression plasmids and miR-NC/miR-942-3p plasmids following treatment with cetuximab for 48 h. CaCO2 $p < 0.0001$, SNU-C1 $p = 0.0002$ (student's $t$-test). (D). Western immunoblotting of apoptosis-related genes (Bax, cleaved Caspase 3, and cleaved Caspase 9) in CaCO2 and SNU-C1 cells transfected with circ-EV/circ-EGFR-overexpression plasmids and miR-NC/miR-942-3p plasmids following treatment with cetuximab for 48 h. GAPDH protein was used as an internal control. Data were representative of at least three independent biological replicates. Bars represent the mean, and error bars indicate the SD. $*p < 0.05$, $**p < 0.01$, $***p < 0.001$, $****p < 0.0001$, ns not significant. Source data are available online for this figure.

the full spectrum of its efficacy has been limited due to the development of therapeutic resistance in a large majority of patients with mCRC (Bardelli and Siena, 2010; Bokemeyer et al, 2012; Misale et al, 2012). Therefore, identifying specific subsets of patients who stand to benefit the most from cetuximab-based therapy could maximize its clinical efficacy and spare the rest from its toxicity and expense. Given that cetuximab is generally offered to patients with WT *KRAS* genes, it is imperative to develop precise predictors of response to cetuximab sensitivity in such patients with mCRC.

Increasing evidence suggests that circRNAs play crucial roles in cancer development and progression (Chen and Shan, 2021; Kristensen et al, 2022; Li et al, 2020). For instance, circ-ANAPC7 suppresses cell proliferation via the PHLPP2-AKT-TGF-β axis in pancreatic cancer (Shi et al, 2022). Likewise, circFNDC3B promotes vasculature formation and metastasis by sponging miR-181c-5p and upregulating SERPINE1 and PROX1 in oral squamous cell carcinoma (Li et al, 2023). However, the role of circRNAs in mediating cetuximab resistance in mCRC remains unexplored. A recent discovery has established that circ-EGFR deprivation not only suppresses tumorigenicity in brain tumor-initiating cells but also enhances the efficiency of nimotuzumab in the treatment of GBM (Liu et al, 2021), suggesting a compelling link between circ-EGFR and treatment efficacy in cancers. Given these intriguing recent reports, we conducted a systematic and comprehensive study to investigate the role of circ-EGFR in mediating sensitivity to cetuximab in colorectal cancer across various cell cultures and animal models.

An increasing body of research supports the role of circRNAs as sponges for miRNAs in modulating downstream target genes in various malignancies (Hansen et al, 2013). miRNAs have emerged as pivotal biomarkers of therapeutic response, including sensitivity to cetuximab. Notably, miR-31-3p expression has been validated as a robust predictive biomarker for cetuximab efficacy in RAS WT mCRC (Laurent-Puig et al, 2019), while miR-100 and miR-125b have been implicated in conferring cetuximab resistance in CRC (Lu et al, 2017). Additionally, Xu et al, developed a miRNA-based assay to predict clinical outcomes in mCRC patients prior to anti-EGFR treatment (Xu et al, 2025). In this study, we successfully identified miR-942-3p as a primary target of circ-EGFR in CRC. We undertook a series of pull-down assays, luciferase reporter assays, and rescue experiments to functionally confirm that circ-EGFR is indeed a miR-942-3p sponge and is a key mediator in determining the efficacy of cetuximab in CRC cells.

GAS1, a known cell growth suppressor linked to the outer membrane via glycosylphosphatidylinositol, was determined as a key regulator of tumorigenesis and cancer progression (Del Sal

et al, 1994; Mo et al, 2016). In this study, we demonstrated that miR-942-3p downregulates the expression of GAS1, while the overexpression of circ-EGFR causes its upregulation in CRC. Furthermore, luciferase reporter assays validated that GAS1 is a direct target of miR-942-3p. Notably, GAS1 is suggested to play a pivotal role in regulating the Hh signaling pathway (Lee et al, 2001). Mounting evidence highlights the critical interplay between the EGFR and Hh pathways in driving tumorigenesis and therapeutic resistance across multiple cancer types (Keysar et al, 2013; Schnidar et al, 2009). Our results, for the first time, identified that the circ-EGFR/miR-942-3p/GAS1/Hh axis is a significant player in enhancing cetuximab sensitivity in CRC. Based on these insights, we reasonably infer that Hh pathway inhibitors and EGFR-targeting antibodies may act synergistically, with their combination potentially delaying or even preventing tumor recurrence.

In our clinical cohort of 45 mCRC patients, we observed an elevated expression of circ-EGFR in the cetuximab-response group, and mCRC patients with high circ-EGFR exhibited a more favorable PFS following cetuximab treatment. Subsequent cell culture experiments and animal models further proved that circ-EGFR sensitizes the response to cetuximab in CRC. Notably, we translated our tissue-based circ-EGFR into a liquid biopsy assay in an independent cohort, achieving an AUC value of 76.9% in predicting anti-EGFR treatment response. Moreover, the positive correlation between circ-EGFR expression in plasma and MTS further reinforces its role as a biomarker for treatment efficacy. This circ-EGFR-based approach effectively reduced the rate of over-treatment without increasing the risk of undertreatment, which is a critical balance in improving patient outcomes while minimizing unnecessary toxicities. Collectively, our findings indicate the clinical potential of circ-EGFR as a predictive biomarker in both tissue and liquid biopsies for mCRC. However, we observed a discrepancy in the distribution of circ-EGFR expression between tissue and liquid biopsy cohorts, which may be attributed to differences in sample origin, cohort size, and the distribution of clinical responses. Therefore, further validation in larger, diverse patient cohorts will be crucial for translating this approach into clinical practice.

In summary, we demonstrate that circ-EGFR is intricately involved in the efficacy of cetuximab through its regulation of the miR-942-3p/GAS1/Hh signaling axis in CRC. Our findings provide comprehensive evidence that circ-EGFR is a promising therapeutic predictor in patients with mCRC, which could potentially inform patient selection and enhance personalized management strategies, particularly for patients with low circ-EGFR expression and who may not benefit from anti-EGFR treatment.

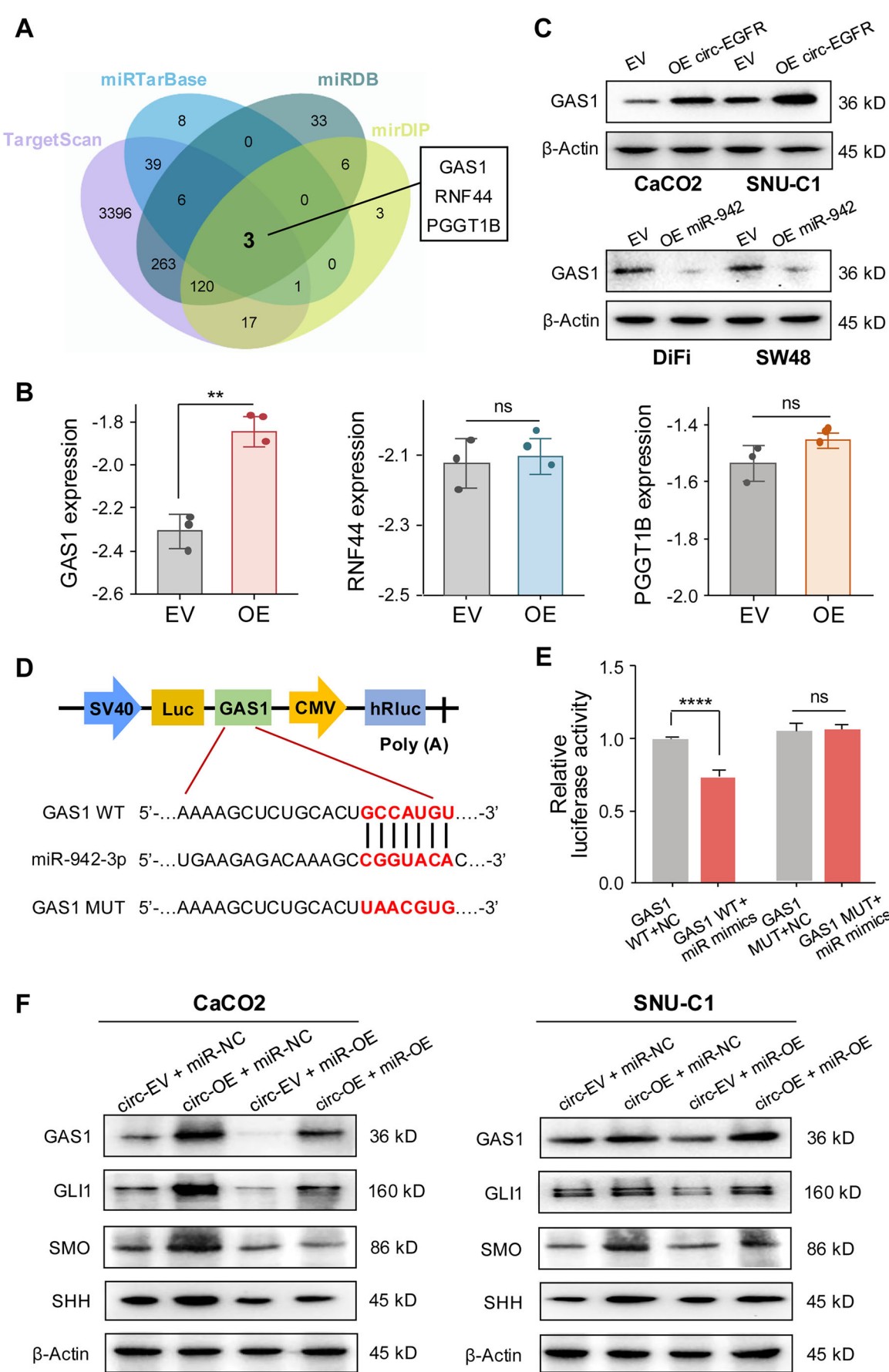

**Figure 6. GAS1 is a direct target of miR-942-3p and is regulated by circ-EGFR in CRC.**

(A) Venn diagram showing the candidate mRNAs predicted to be the binding targets of miR-942-3p by TargetScan, miRTarBase, miRDB, and mirDIP. (B) Relative expression of three candidate mRNAs after circ-EGFR overexpression. GAS1 $p = 0.002$, RNF44 $p = 0.711$, PGGT1B $p = 0.115$ (student's *t*-test). (C) GAS1 protein levels in CaCO2 and SNU-C1 cells transfected with circ-EV and circ-EGFR-overexpressing plasmids (upper panel). GAS1 protein levels in DiFi and SW48 cells transfected with NC and miR-942-3p mimics (lower panel). (D) Schematic illustration of GAS1 WT and MUT luciferase reporter vectors were shown. (E) The luciferase activity of GAS1 WT or GAS1 MUT after transfection with miR-942-3p mimics in 293T cells. WT $p < 0.0001$, MUT $p = 0.822$ (student's *t*-test). (F) The relative protein levels of GAS1 and hedgehog (Hh) regulators were detected by western blotting in miR-942-3p rescue experiments. Data were representative of at least three independent biological replicates. Bars represent the mean, and error bars indicate SD. *$p < 0.05$, **$p < 0.01$, ***$p < 0.001$, ****$p < 0.0001$, ns not significant. Source data are available online for this figure.

# Methods

## Reagents and tools table

| Reagent/resource | Reference or source | Identifier or catalog number |
|---|---|---|
| RIP Lysis Buffer | Sigma, St. Louis, MO, USA | R0278 |
| Protease Inhibitor Cocktail | Thermo Fisher Scientific | 78429 |
| RNase Inhibitors | Thermo Fisher Scientific | N8080119 |
| Dynabeads™ Protein G for Immunoprecipitation | Invitrogen | 10-004-D |
| QIAzol Lysis Reagent | Qiagen, Hilden, Germany | 79306 |
| Biotin-labeled circ-EGFR probes (custom-synthesized) | Servicebio, Wuhan, China | N/A |
| Pull-down lysis buffer | Thermo Fisher Scientific | PI87787 |
| Streptavidin Magnetic Beads | Thermo Fisher Scientific | 88817 |
| Luciferase reporter vector (PGL3-CMV-LUC-MCS) | Addgene | RRID: Addgene_17186 |
| miR-942-3p mimics and negative control mimics | GenePharma, Shanghai, China | custom synthesized |
| Dual-Luciferase® Reporter Assay System | Promega | E1910 |
| DMEM (Dulbecco's Modified Eagle Medium) | Gibco (Thermo Fisher Scientific) | 11965092 |
| RPMI-1640 Medium | Gibco (Thermo Fisher Scientific) | 11875093 |
| Fetal bovine serum (FBS) | Gibco (Thermo Fisher Scientific) | A5256701 |
| Penicillin-Streptomycin Solution | Gibco (Thermo Fisher Scientific) | 15140122 |
| Third-generation packaging system mix & lentifectin combo | Applied Biological Materials (ABM), Canada | LV053 & G074 |
| Puromycin | Thermo Fisher Scientific | A1113802 |
| AllPrep DNA/RNA FFPE Kit | Qiagen, Hilden, Germany | 80234 |
| RNeasy Mini Kit | Qiagen | 74104 |
| miRNeasy Serum/Plasma Kit | Qiagen | 217184 |

| Reagent/resource | Reference or source | Identifier or catalog number |
|---|---|---|
| High-capacity cDNA reverse transcription Kit | Applied Biosystems, Foster City, CA, USA | 4368814 |
| miRCURY LNA RT Kit | Qiagen | 339340 |
| SensiFAST SYBR LO-ROX Kit | Bioline, London, UK | Cat# BIO-94005 |
| BCA Protein Assay Kit | Thermo Fisher Scientific | 23227 |
| PVDF Membrane | Cytiva (formerly GE Healthcare), USA | GE10600029 |
| TBS reagent | Bio-Rad | 1706435 |
| Tween 20 | Sigma-Aldrich | P1379 |
| HRP Chemiluminescent Substrate Kit | Thermo Fisher Scientific | 32106 |
| RNase-R enzyme | Epicentre Technologies | RNR07250 |
| MTT | Sigma-Aldrich | 475989 |
| DMSO | Sigma-Aldrich | D2650 |
| Phosphate-buffered saline (PBS) | Thermo Fisher Scientific | 10010023 |
| Matrigel | BD Biosciences | 356234 |
| Diff-Quik Staining Kit | Medix | B41331 |
| Muse Annexin V & Dead Cell Reagent | Luminex Corporation, Austin, TX | MCH100105 |
| FISH Detection Kit | Servicebio, Wuhan, China | N/A |
| Proteinase K | Thermo Fisher Scientific | FEREO0491 |
| DAPI mounting medium | MedChemExpress | HY-D0822 |
| SSC buffer | Thermo Fisher Scientific | J60839 |
| **Experimental models** | | |
| BALB/c nude mice (male, 5 weeks, 18–20 g) | Charles River Laboratories (USA) | *RRID: IMSR_CRL:194* |
| CaCO2 cell line | ATCC | RRID: CVCL_0025 |
| SNU-C1 cell line | ATCC | RRID: CVCL_1708 |
| HT29 cell line | ATCC | RRID: CVCL_A8EZ |
| HEK-293T cell line | ATCC | RRID: CVCL_0063 |
| SW48 cell line | Dr. Wenyi Wei, Harvard Med. School | RRID: CVCL_1724 |
| DiFi cell line | Dr. Wenyi Wei, Harvard Med. School | RRID: CVCL_6895 |

| Reagent/resource | Reference or source | Identifier or catalog number |
|---|---|---|
| FFPE tissue specimens from mCRC patients | GEMCAD 17-01 trial, Spain | N/A |
| Plasma samples from mCRC patients (liquid biopsy) | PULSE and POSIBA trials | PULSE: NCT01288339; POSIBA: NCT01276379 |
| **Recombinant DNA** | | |
| pcDNA 3.1 plasmid (empty vector) | Addgene | RRID: Addgene_18951 |
| circ-EGFR overexpression plasmid (pcDNA 3.1 backbone) | This paper | custom construct |
| hsa-miR-942-3p overexpression plasmid | This paper | custom construct |
| Lv-CMV-GFP vector | Genepharma, Shanghai, China | RRID: Addgene_17448 |
| **Antibodies** | | |
| GAS1 | Proteintech Group Inc, IL, USA | 17903-1-AP |
| Bax | Cell Signaling Technology | 5023 s |
| Cleaved Caspase 3 | Cell Signaling Technology | 9661 s |
| Cleaved Caspase 9 | Cell Signaling Technology | 9509 |
| SHH | Proteintech | 20697-1-AP |
| SMO | Proteintech | 66851-1-Ig |
| GLI1 | Proteintech | 66905-1-Ig |
| GAPDH | Proteintech | 10494-1-AP |
| β-Actin | Cell Signaling Technology | 58169 s |
| HRP-conjugated anti-rabbit secondary antibody | Cell Signaling Technology | 7074 |
| HRP-conjugated anti-mouse secondary antibody | Cell Signaling Technology | 7076 |
| **Oligonucleotides and other sequence-based reagents** | | |
| β-actin primer | Integrated DNA Technologies, Coralville, IA, USA | Custom construct |
| EGFR primer | Integrated DNA Technologies, Coralville, IA, USA | Custom construct |
| GAS1 primer | Integrated DNA Technologies, Coralville, IA, USA | Custom construct |
| PGGT1B primer | Integrated DNA Technologies, Coralville, IA, USA | custom construct |
| RNF44 primer | Integrated DNA Technologies, Coralville, IA, USA | Custom construct |
| circ-EGFR primer | Integrated DNA Technologies, Coralville, IA, USA | Custom construct |
| hsa-miR-942-3p primer | Qiagen | ZP00002559 |

| Reagent/resource | Reference or source | Identifier or catalog number |
|---|---|---|
| hsa-miR-665 primer | Qiagen | YP00204710 |
| hsa-miR-609 primer | Qiagen | YP00204613 |
| hsa-miR-296-5p primer | Qiagen | YP00204436 |
| hsa-miR-1184 primer | Qiagen | YP00204330 |
| miR-16-5p primer | Qiagen | YP00205702 |
| **Chemicals, enzymes and other reagents** | | |
| Cetuximab | MedChemExpress | HY-P9905 |
| Sonidegib | MedChemExpress | HY-15152 |
| Vismodegib | MedChemExpress | HY-10440 |
| Methanol | Thermo Fisher Scientific | A412-4 |
| Crystal violet | Thermo Fisher Scientific | C581-25 |
| **Software** | | |
| R (version 4.1.2) | The R Project | RRID:SCR_001905 |
| GraphPad Prism (v9) | GraphPad Software | RRID:SCR_002798 |
| QuantStudio 7 Flex Real-Time PCR Software | Thermo Fisher Scientific | RRID:SCR_020245 |
| ImageJ Software | N/A | RRID: SCR_003070 |
| Zen (for Zeiss microscope) | Carl Zeiss Microscopy | RRID:SCR_013672 |
| **Other** | | |

## Patient specimens

A total of 45 formalin-fixed paraffin-embedded (FFPE) tissue specimens were obtained from mCRC patients enrolled in the GEMCAD 17-01 trial, an open-label, randomized phase II trial, at several hospitals in Spain. These patients were genetically characterized for a *KRAS* status, and all mCRC patients with the wild-type gene received first-line treatment with chemotherapy plus cetuximab. Tumor specimens were collected prior to treatment initiation, and all available samples from patients who received anti-EGFR therapy in the trial were included in this study.

For the liquid biopsy cohort, 112 patient samples were obtained from two previously described trials, PULSE and POSIBA, as reported in prior publications (Alonso et al, 2018; Garcia-Albeniz et al, 2019; Maurel et al, 2019). The PULSE and POSIBA trials are open-label, multicenter studies conducted nationwide. The inclusion criteria of these trials required adult patients with unresectable RAS-wild-type metastatic colorectal cancer who had at least one lesion evaluable by RECIST criteria, were in good systemic health (ECOG performance status 0–1), and had adequate liver, kidney, and bone marrow functions. Patients received first-line anti-EGFR inhibitors combined with chemotherapy, irrespective of the primary tumor location (panitumumab or cetuximab, respectively). The exclusion criteria included prior anti-EGFR therapy, uncontrolled comorbidities, or other conditions that could interfere with study procedures. All three parent trials (GEMCAD 17-01, PULSE, and POSIBA) were open-label studies without blinding.

Human samples were obtained from study participants with a mean age of 68.2 ± 4.5 years, 58.7% of whom were male. The study population was predominantly Caucasian. All experiments

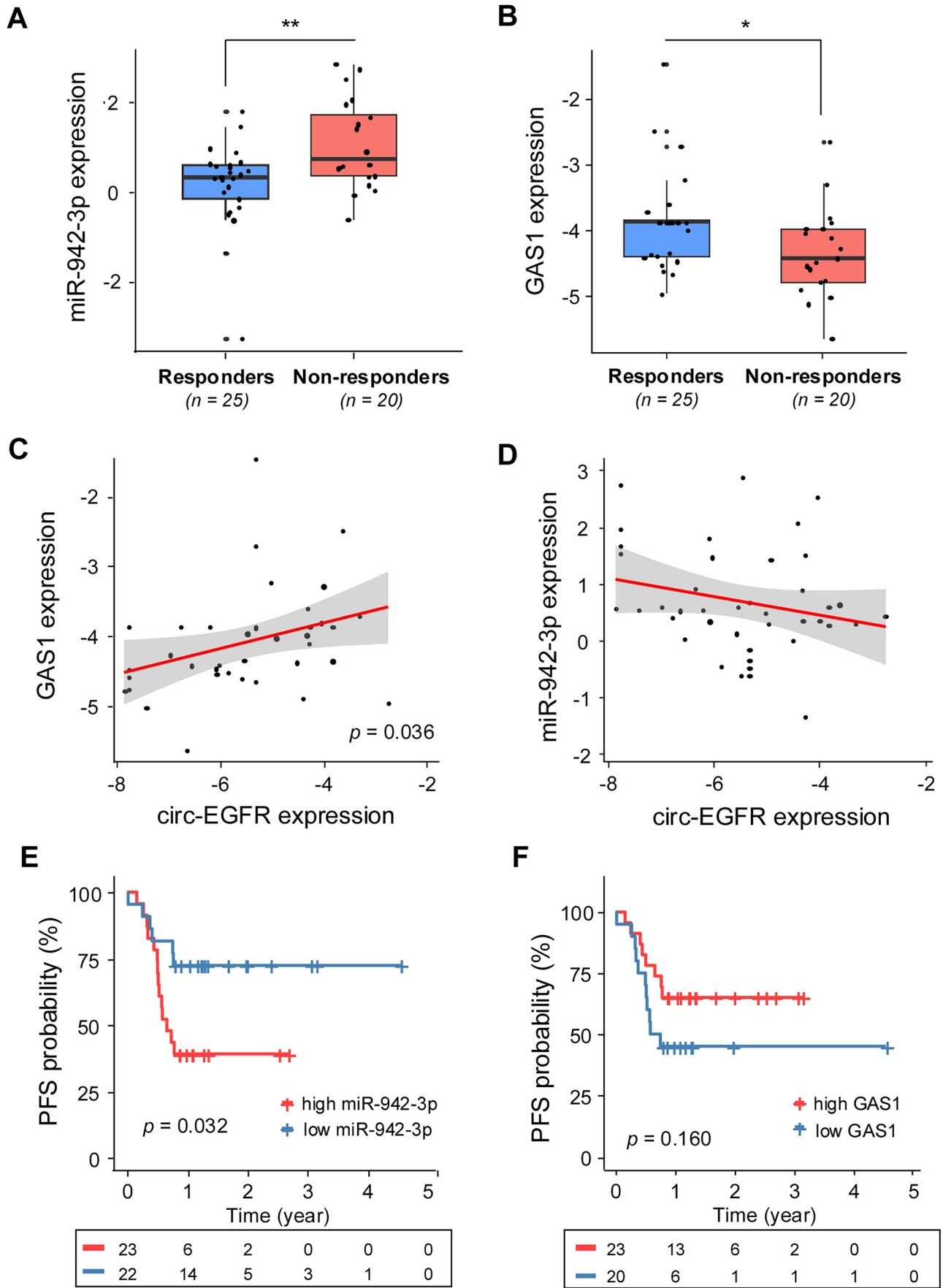

**Figure 7. Circ-EGFR is a potential predictive biomarker for treatment response in clinical practice.**

(A, B) The relative expression level of miR-942-3p (A) and GAS1 (B) was evaluated by RT-qPCR in a cohort of 45 cases with varying responses to cetuximab treatment. Boxplot (P25 − 1.5 ×IQR; P25; median; P75; P75 + 1.5×IQR, Tukey method). miR-942-3p expression in responders group: minimum = −3.263, maximum = 1.795, median = 0.346, whisker low = −1.281, whisker high = 1.725; non-responders: minimum = −0.620, maximum = 2.865, median = 0.754, whisker low = −0.620, whisker high = 2.865; Responders vs. non-responders: $p = 0.003$; GAS1 expression in responders group: minimum = −4.954, maximum = −1.463, median = −3.866, whisker low = −4.953, whisker high = −2.997; non-responders: minimum = −5.632, maximum = −2.641, median = −4.419, whisker low = −5.632, whisker high = −2.741; Responders vs. non-responders: $p = 0.047$ (student's $t$-test). (C) The correlation between circ-EGFR expression and miR-942-3p expression in the clinical cohort. $p = 0.036$ (Pearson correlation analysis). (D) The correlation between circ-EGFR expression and GAS1 expression in the clinical cohort. $p = 0.138$ (Pearson correlation analysis). (E, F). Kaplan–Meier analysis showed correlations between miR-942-3p (E) or GAS1 (F) expression and prognoses of CRC patients in the clinical cohort. miR-942-3p $p = 0.032$, GAS1 $p = 0.160$ (log-rank test). *$p < 0.05$, **$p < 0.01$, ***$p < 0.001$, **** $p < 0.0001$, ns not significant. Source data are available online for this figure.

involving human participants were conducted in accordance with the ethical principles outlined in the World Medical Association (WMA) Declaration of Helsinki and the US Department of Health and Human Services Belmont Report. All participants provided informed written consent, and the Institutional Review Board of Beckman Research Institute approved the study (IRB 23228). The trials are registered and completed on ClinicalTrials.gov (PULSE NCT01288339 and POSIBA NCT01276379). As this was a retrospective study, the sample size was determined based on specimen availability from the completed clinical trials.

## Cell culture and reagents

Human colorectal cancer cell lines, CaCO2 (RRID:CVCL_0025), SNU-C1 (RRID:CVCL_1708), HT29 (RRID:CVCL_A8EZ), and 293T (RRID:CVCL_0063) cells were purchased from the American Type Culture Collection (ATCC), while SW48 (RRID:CVCL_1724) and DiFi (RRID:CVCL_6895) were generously provided by Dr. Wenyi Wei from Harvard Medical School, Boston, MA, USA (Liu et al, 2021). Additionally, all cell lines were routinely tested for mycoplasma contamination using MycoAlert Mycoplasma Detection Kit and confirmed to be mycoplasma-free throughout the study. CaCO2, SNU-C1, HT29, and 293T cells were cultured in Dulbecco's modified Eagle's medium (DMEM; Gibco, Carlsbad, CA, USA) supplemented with 10 FBS and 1% penicillin-streptomycin. SW48 and DiFi cell lines were cultured in RPMI-1640 medium (Gibco, Carlsbad, CA, USA) containing 10% fetal bovine serum (Gibco) and 1% penicillin-streptomycin (Gibco). All cells were maintained in a temperature-controlled incubator at 37 °C with a 5% humidified $CO_2$ atmosphere, and the cell culture medium was refreshed every two days.

## Construction of stable overexpressing and knockdown CRC cell lines

For the construction of circ-EGFR overexpression (OE) plasmid, hsa_circ_0080230 was constructed into pcDNA 3.1 plasmid (Invitrogen, Carlsbad, CA, USA) with an empty vector (EV) as a negative control (pcDNA 3.1, RRID: Addgene_18951). As for miR-942-3p plasmid, hsa-miR-942-3p overexpressing lentiviral constructs were generated using synthetic oligonucleotides and Lv-CMV-GPF vector (Genepharma, Shanghai, China, RRID: Addgene_17448). To construct stable overexpression cell lines, CaCO2 and SNU-C1 cells were transduced with circ-EGFR, miRNA plasmid, or an EV (pcDNA 3.1 or Lv-CMV-GPF) using third third-generation packaging system mix & lentifectin combo (Applied Biological Materials, Richmond, BC, Canada), followed by

puromycin selection for 7 days (4 µg/mL). Puromycin-resistant clones were isolated and subsequently screened for gene expression using reverse-transcription quantitative polymerase chain reaction (RT-qPCR) assays.

For the depletion of circ-EGFR in CRC cell lines, two specific small hairpin RNAs (shRNAs) targeted against circ-EGFR were cloned into an LV3 lentiviral vector to achieve a stable knockdown of the target genes in various CRC cells. Lentiviral particles were generated using HEK-293T cells and the 3rd Generation Packaging Mix & Lentifectin Combo (Applied Biological Materials). Lentivirus packaging and lentivirus infection were conducted according to the manufacturer's protocol. To generate stable knockdown of circ-EGFR in cell lines, SW48 and DiFi cells transfected with sh-circ-EGFR or non-target control lentivirus were selected by puromycin for 7 days (2 µg/mL). All oligonucleotides used in this study are presented in Table EV2.

## Genomic DNA extraction

Genomic DNA (gDNA) was purified using an AllPrep DNA/RNA FFPE Kit (Qiagen, Hilden, Germany). The integrity and purity of extracted total gDNA were detected using a NanoDrop system (Thermo Fisher Scientific, Waltham, MA, USA). The DNA products were immediately stored at -80 °C until future use.

## RNA extraction and RT-qPCR assays

Total RNA was extracted from frozen specimens and cell lines using the RNeasy mini kit (Qiagen) and from blood samples using the miRNeasy Serum/Plasma Kit (Qiagen). The integrity and purity of extracted total RNA were detected using a NanoDrop system (Thermo Fisher Scientific), and 500-1000 ng RNA was used for complementary DNA (cDNA) synthesis. For circRNA and mRNA, cDNA was synthesized using the High-Capacity cDNA Reverse Transcription Kit with an RNase inhibitor (Applied Biosystems, Foster City, CA, USA). For all circ-EGFR isoforms, specific divergent primers were designed using the widely recognized CircInteractome web tool (Dudekula et al, 2016). To ensure primer specificity for circRNAs, unique primer sets were designed targeting the splice-site junctions. For miRNA, cDNA synthesis was performed using the miRCURY LNA RT Kit (Qiagen). The quantification of RNA expression levels was performed using a SensiFAST™ SYBR® LO-ROX Kit (Bioline, London, UK) on a QuantStudio 7 flex real-time quantitative PCR system (Applied Biosystems, Foster City, CA, USA, RRID:SCR_020245). β-actin was used as an endogenous control for quantifying the expression of circRNAs and mRNAs. Following an assessment of endogenous control miRNAs documented in the literature (e.g., miR-16-5p,

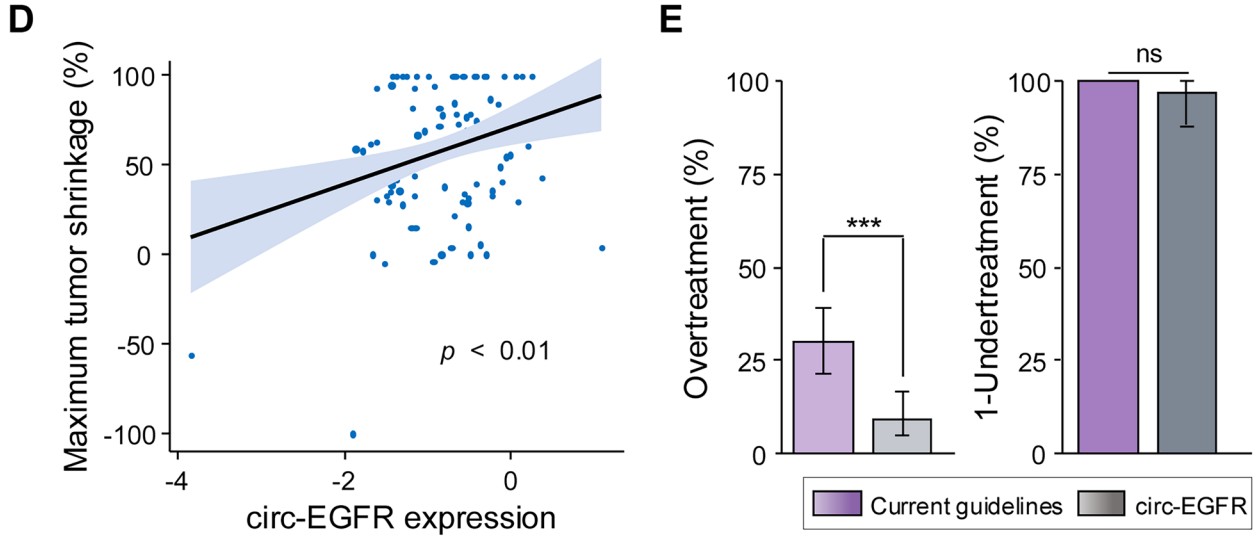

◄ 

◄ **Figure 8.** Liquid biopsy circ-EGFR serves as a promising predictive biomarker for cetuximab response in CRC.

RNU6B, miR-39-3p, and miR-30a-5p), miR-16-5p was selected as the optimal control miRNA, based on its stable expression across all samples. The expression values of RNAs were calculated by the $2^{-\Delta CT}$ method. The primers used in this study are listed in Table EV3.

## Western immunoblotting

Colorectal cancer cells were lysed with cold RIPA lysis buffer (Thermo Fisher Scientific) containing a 1% protease inhibitor cocktail (Thermo Fisher Scientific). Afterward, the lysates were centrifuged at $15,000 \times g$ for 15 min at 4 °C. Total protein concentration was quantified with a BCA Protein Assay Kit (Thermo Fisher Scientific). Total proteins were loaded on an SDS gel and transferred to an activated polyvinylidene fluoride film (PVDF) membrane (Cytiva, Marlborough, MA, USA). PVDF membranes were blocked in 5% non-fat dry milk prepared in TBST for 90 min and incubated with primary antibodies overnight at 4 °C. After extensive washing with TBST, the membranes were incubated with horseradish peroxidase-coupled secondary antibodies (#7074 RRID: AB_2099233 or #7076 RRID: AB_330924; Cell Signaling Technology, Danvers, MA, USA) for 60 min. The immunoblots were subsequently exposed to an HRP-based chemiluminescence kit (Thermo Fisher Scientific), and the signal was imaged using an electrochemiluminescence (ECL) device (Bio-Rad, Hercules, CA, USA, SCR_019037). The list of primary antibodies is described in Table EV4. The protein expressions were quantified using the ImageJ software, and subsequently, the relative changes were determined.

## RNase-R treatment

RNase-R (Epicentre Technologies, Madison, Wisconsin, USA) was used to digest linear RNAs in this study. Approximately 2 ug of RNA sample was divided into two equal parts: the RNase-R+ group received 0.2 μL RNase-R along with 0.6 μL buffer, while the RNase-R- group received 0.2 μL DEPC-H$_2$O along with 0.6 μL buffer. The samples were incubated at 37 °C for 30 min.

## Cell viability and colony formation assays

To perform cell viability assays, various cell lines were seeded into 96-well plates at a density of 3000 cells/well. After 24 h of cell seeding to facilitate cell adherence to the plates, cells were treated with cetuximab at concentrations of 0, 5, 10, 20, 40, 80, and 160 μg/mL. After 48-h treatment, the cells were incubated with a fresh culture medium containing 3-(4,5-dimethyl-2-thiazolyl)-2,5-diphenyl-2H-tetrazolium bromide (MTT, 5 mg/mL) for 4 h. Subsequently, the MTT

reagent was removed, and 100 μL of dimethyl sulfoxide (DMSO) was added to dissolve the methanogenic crystals at 37 °C. The absorbance of the cells was measured at 550 nm using an enzymatic plate analyzer (Molecular Devices, San Jose, CA, USA). The half maximal inhibitory concentration (IC$_{50}$) of cetuximab was defined as the drug concentration required to inhibit CRC cells by 50% relative to controls.

Cells were seeded at a density of 1000 cells/well for colony formation assays in six-well plates. The cells were maintained in a complete medium for 10–14 days, and the medium was replenished every 3 days. The colonies were fixed with methanol for 15 min and stained with 0.1% crystal violet (Thermo Fisher Scientific) overnight. After washing with PBS, the number of colonies was counted. Data were derived from a minimum of three independent experiments.

## Wound healing and transwell invasion assays

For wound healing assays, CRC cells were treated with cetuximab at respective IC$_{50}$ concentrations (SW48: 35 μg/mL, DiFi: 4 μg/mL) for 48 h. Subsequently, cells were seeded into 6-well plates to ensure they reached 80–90% confluency the following day. Wounds (i.e., scratches) were created by scraping the cell monolayers, and the wound-healing potential of the cells was observed under a microscope at a magnification of ×40.

For the invasion assays, $5 \times 10^4$ cells were cultured in 24-well transwell chambers with 8-μm pore size, pre-coated with Matrigel (BD Biosciences, Franklin Lakes, NJ, USA). After 48 h, Diff-Quik staining was used to detect invaded cells.

## Apoptosis assays

CRC cells were treated with Cetuximab at IC$_{50}$ concentrations (SW48: 35 μg/mL, DiFi: 4 μg/mL, CaCO$_2$: 160 μg/mL, SNU-C1: 160 μg/mL) for 48 h. After that, 100 μL of Muse Annexin V & Dead Cell Reagent (Luminex Corp, Austin, TX, USA) was added to the 100 μL of cell suspension. The proportion of apoptotic cells was detected using the Muse Cell Analyzer (EMD Millipore Corp, Hayward, CA, USA, RRID:SCR_020252) according to the manufacturer's instructions.

## Synergy assessment using SynergyFinder

CaCO2 and SNU-C1 cells were seeded into 96-well plates and treated with the compounds as described below. Treatments included single agents (Cetuximab, Sonidegib, and Vismodegib) as well as combination therapies (Cetuximab with Sonidegib, or Cetuximab with Vismodegib). Drug concentration gradients were determined based on the individual IC$_{50}$ values, with Cetuximab,

Sonidegib, and Vismodegib administered at concentrations of 0, 0.01, 0.1, 1, and 10 μM. After 48 h of treatment, cell viability was assessed by measuring absorbance at 550 nm using an enzymatic plate analyzer. Drug synergy scores were calculated using the SynergyFinder online software (https://synergyfinder.fimm.fi), based on the response surface model and the Zero Interaction Potency (ZIP) method. ZIP synergy scores above 0 indicated positive drug interaction, while scores greater than 10 reflected strong synergy and were marked in red on the response surface plots.

## In vivo mice model

Male immunodeficient BALB/c nude mice (five weeks old and weighing 18–20 g) were purchased from Charles River (Wilmington, MA, USA, RRID:IMSR_CRL:194). Mice were housed in a pathogen-free facility under controlled environmental conditions (12-h light/12-h dark cycle, temperature 20–24 °C, relative humidity 40–60%) with free access to standard chow diet and water. Animals were monitored daily by trained staff, and humane endpoints were applied according to approved protocols. All animal experiments were performed in accordance with institutional guidelines and approved by the Institutional Animal Care and Use Committee (IACUC) of City of Hope, under protocol number (IRB 23228). To generate the xenograft models, $5 \times 10^6$ HT29 cells with stable overexpression of circ-EGFR ($n = 10$), or mock controls ($n = 10$), were subcutaneously inserted into the right flank of the mouse using a 27-gauge needle. When the tumors became visibly apparent (about 1 week after the cell injection), we treated nude mice bearing xenograft tumors with either saline or cetuximab (40 mg/kg per mouse, biweekly) for 2 weeks. The tumor size and body weights of the mice were measured and recorded every other day. After ~21 days of subcutaneous injection, the animals were euthanized, and each tumor was carefully dissected, weighed, and promptly frozen in liquid nitrogen.

To establish a mouse model of CRC liver metastasis, $5 \times 10^6$ HT29 cells (including mock controls and circ-EGFR overexpression) were injected into the spleens of male BALB/c nude mice purchased from Charles River (Wilmington, MA, USA, RRID:IMSR_CRL:194). The specific procedure was as follows: mice were first anesthetized with 3% isoflurane for induction and maintained on 1.5–2% isoflurane during surgery using a precision vaporizer. Once fully anesthetized, the surgical area was disinfected, and a ~5 mm longitudinal incision was made along the left axillary line, just below the posterior margin of the rib cage. The spleen was carefully exteriorized from the abdominal cavity, and a single-cell suspension containing $5 \times 10^6$ HT29 cells was promptly injected into its lower pole. The injection site was then gently pressed with an alcohol swab to prevent leakage and ensure hemostasis. After confirming the absence of active bleeding, the abdominal wall and skin were closed using interrupted sutures. Liver metastases were monitored by in vivo bioluminescence imaging on day 7 post-injection. Upon confirmation of liver metastasis formation, mice were treated with either saline or cetuximab (40 mg/kg per mouse, administered biweekly) for 2 weeks. At the end of the treatment period (approximately day 21), bioluminescence imaging was performed to evaluate tumor burden. Mice were randomly divided into four groups: (1) Ctrl + Saline: mock control HT29 cells treated with saline; (2) OE + Saline: circ-EGFR overexpressing HT29

treated with saline; (3) Ctrl + Cet: mock control HT29 treated with cetuximab; (4) OE + Cet: circ-EGFR overexpressing HT29 treated with cetuximab. On day 30, all mice were sacrificed, and tumor and liver tissues were collected for histopathological analysis. All experiments were performed in strict accordance with the guidelines outlined in the 'Guide for the Care and Use of Laboratory Animals' published by the NIH. The Institutional Animal Care and Use Committee (IACUC) of the City of Hope duly approved the study protocol.

## RNA fluorescence in situ hybridization (FISH)

The circ-EGFR and miR-942-3p probes were designed and synthesized by Servicebio. The FISH assay for tissue samples and cells was conducted according to the manufacturer's instructions using the FISH detection kit (Servicebio). Briefly, FFPE tissue sections were deparaffinized in xylene twice for 15 min each, followed by rehydration in graded ethanol (100, 85, and 75%) for 5 min per concentration. Antigen retrieval was then performed at 100 °C for 15 min in 20 mM sodium citrate buffer containing Tween 20 and trisodium citrate dihydrate, after which the sections were cooled to room temperature. The tissue sections were then permeabilized with 0.1% Triton X-100 in PBS for 30 min. For cells, fixation was performed using paraformaldehyde for 15 min, followed by membrane permeabilization with 0.1% Triton X-100 in PBS. The cells were digested using 100 μL proteinase K at 37 °C for 10 min. Subsequently, both tissue sections and cells were pre-hybridized with the hybridization solution at 40 °C for 30 min, followed by hybridization with multi-target probes 1 and 2 for 3 h and 45 min, respectively. After washing with SSC, a final hybridization was performed using a multi-fluorescence signal probe. The slides were then mounted with a DAPI-containing mounting medium and washed with SSC. The sections were observed under a confocal laser scanning microscope (Carl Zeiss, Göschwitzer Strasse, Germany, SCR_020925), with the fluorescence signal detected using the appropriate fluorescence detection channel or filter under a 40× - 63× objective.

## RNA immunoprecipitation (RIP)

After centrifugation, 1 ml of RIPA lysis Buffer (Sigma, St. Louis, Missouri, USA) containing a protease inhibitor cocktail (Thermo Fisher Scientific) and RNase inhibitors (Thermo Fisher Scientific) was added to the cell pellet. The lysate supernatant was divided into two parts and incubated with an AGO2 antibody (Proteintech, Rosemont, IL, USA) and Dynabeads™ Protein G beads for Immunoprecipitation (Invitrogen), followed by uniform rotation at 4 °C overnight. Total RNAs were purified and extracted from bead complexes using QIAzol Lysis Reagent (Qiagen, Hilden, Germany) for RT-qPCR analysis.

## RNA Pull-down

The circ-EGFR and control probes were designed and synthesized by Servicebio (Wuhan, Hubei, China). About $1 \times 10^7$ cells were harvested and lysed with pull-down lysis buffer. About 50 pmol 3' end biotin-labeled circ-EGFR probes were bound with 50 ul streptavidin magnetic beads (Thermo Fisher Scientific) in RNA capture buffer for 1 h at room temperature with agitation.

**The Paper Explained**

**Problem**

Approximately 50% of colorectal cancer (CRC) patients develop metastasis, with 20–25% presenting with metastatic colorectal cancer (mCRC) at initial diagnosis. Cetuximab offers beneficial yet limited clinical improvements for patients with KRAS wild-type (WT) mCRC. The role of circ-EGFR in response to EGFR-targeted therapy in mCRC remains largely unexplored.

**Results**

We have identified circ-EGFR as a promising predictive biomarker for response to cetuximab therapy. Mechanistically, circ-EGFR functions as a sponge for miR-942-3p, resulting in the upregulation of GAS1 and activation of the Hedgehog (Hh) signaling pathway, thereby promoting the efficacy of cetuximab in CRC. Finally, we successfully translated it into a liquid biopsy predictor for anti-EGFR response in mCRC.

**Impact**

By elucidating the role of circ-EGFR in modulating tumor response to cetuximab via the circ-EGFR/miR-942-3p/GAS1/Hh regulatory axis, our findings provide key molecular insights into the biologic changes associated with varying responses to anti-EGFR therapy in CRC. The successful implementation of the circ-EGFR predictive approach is anticipated to inform patient selection, enhance personalized management strategies, and ultimately improve patient outcomes.

Streptavidin beads-conjugated RNA probes were then incubated with 200 mg of cell lysate to pull down at 4 °C overnight. Total RNA was eluted and extracted from probe-bead complexes for RT-qPCR.

## Luciferase reporter assays

The sequences of circ-EGFR and GAS1-3'UTR, along with their corresponding mutant (MUT) counterparts, were designed, synthesized, and subsequently inserted into a luciferase reporter vector (PGL3-CMV-LUC-MCS, RRID: Addgene_17186). These constructs included circ-EGFR-WT (hsa_circ_0080230-WT), circ-EGFR-MUT (hsa_circ_0080230-MUT), GAS1-3'-UTR-WT, and GAS1-3'-UTR-MUT, respectively. These reporter plasmids were co-transfected with miR-942-3p or control mimics into HEK-293T cells, respectively. After that, the relative luciferase activity was evaluated using the Dual-Luciferase Assay Kit (Promega, Madison, Wisconsin, USA).

## Statistical analysis

All statistical analyses were conducted using software R (version 4.1.2, https://www.r-project.org, RRID:SCR_001905) and Graph-Pad Prism (Version 9, RRID: SCR_002798). The Mann-Whitney U or t-test was applied to compare two independent groups with continuous variables. The chi-square test was performed to compare the proportions between the two groups. Kaplan–Meier survival analysis was used to evaluate the associations between gene expression levels and survival outcomes using the "survival" and "survminer" packages in R. The performance evaluation, using ROC curves and Youden's index, was performed with the "pROC" package in R. Pearson tests were conducted for correlation analysis.

A $p$ value of <0.05 was considered a statistically significant change. Our results were derived from a minimum of three independently conducted experiments.

## Data availability

This study includes no data deposited in external repositories.

The source data of this paper are collected in the following database record: biostudies:S-SCDT-10_1038-S44321-025-00333-0.

## Peer review information

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

## Acknowledgements

We would like to express our sincere gratitude to all individuals who contributed to the completion of this study. We thank the Animal Care Program, Auxiliary Services, and the Light Microscopy Core at City of Hope for their valuable technical support in animal handling and imaging. We are

grateful to our collaborators for generously providing samples and clinical information. We also extend our special thanks to Dr. Wenyi Wei for kindly providing the cell lines used in this research. We appreciate Dr. Caiming Xu's support in data analysis. This work was supported by the CA72851, CA181572, CA227602, CA214254, CA271443, and CA187956 grants from the National Cancer Institute, National Institutes of Health.

## Author contributions

**Silei Sui**: Conceptualization; Data curation; Software; Formal analysis; Investigation; Methodology; Writing—original draft. **Yuan Li**: Investigation; Methodology; Writing—original draft. **Joan Maurel**: Resources; Writing—review and editing. **Ajay Goel**: Conceptualization; Supervision; Funding acquisition; Project administration; Writing—review and editing.

Source data underlying figure panels in this paper may have individual authorship assigned. Where available, figure panel/source data authorship is listed in the following database record: biostudies:S-SCDT-10_1038-S44321-025-00333-0.

## Disclosure and competing interests statement

The authors declare no competing interests.

# Expanded View Figures

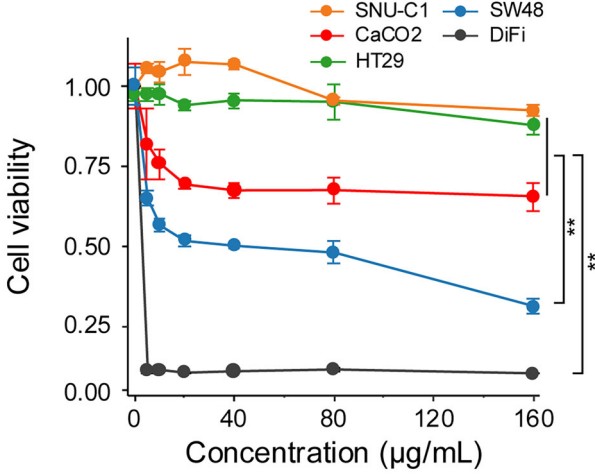

| IC$_{50}$ (µg/mL) | CaCO2 | SNU-C1 | HT29 | SW48 | DiFi |
|---|---|---|---|---|---|
| | N/A | N/A | N/A | 34.77 | < 5 |

**Figure EV1. Cell viability of five colorectal cancer cell lines measured 48 h after initial exposure to cetuximab.**

GEM-R vs. SW48: $p = 0.002$, GEM-R vs. DiFi: $p = 0.001$ (two-way ANOVA). Data were representative of three independent biological replicates. Bars represent the mean, and error bars indicate SD. **$p < 0.01$, N/A not available.

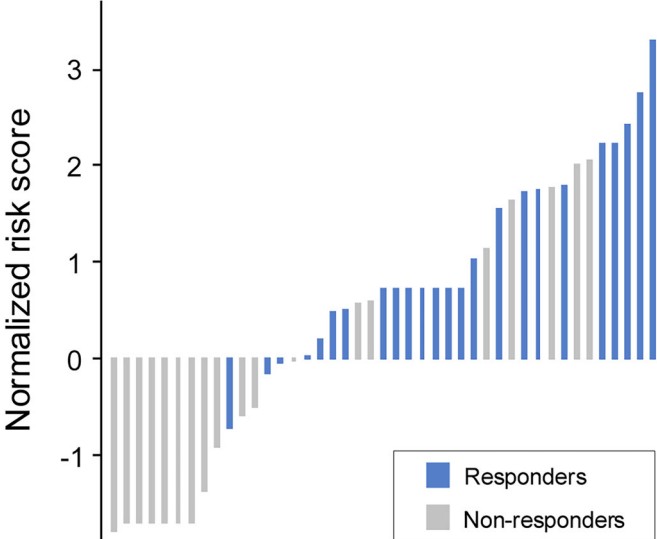

**Figure EV2.** The waterfall plot depicts the risk probability distribution of the normalized risk score between the responder and non-responder groups.

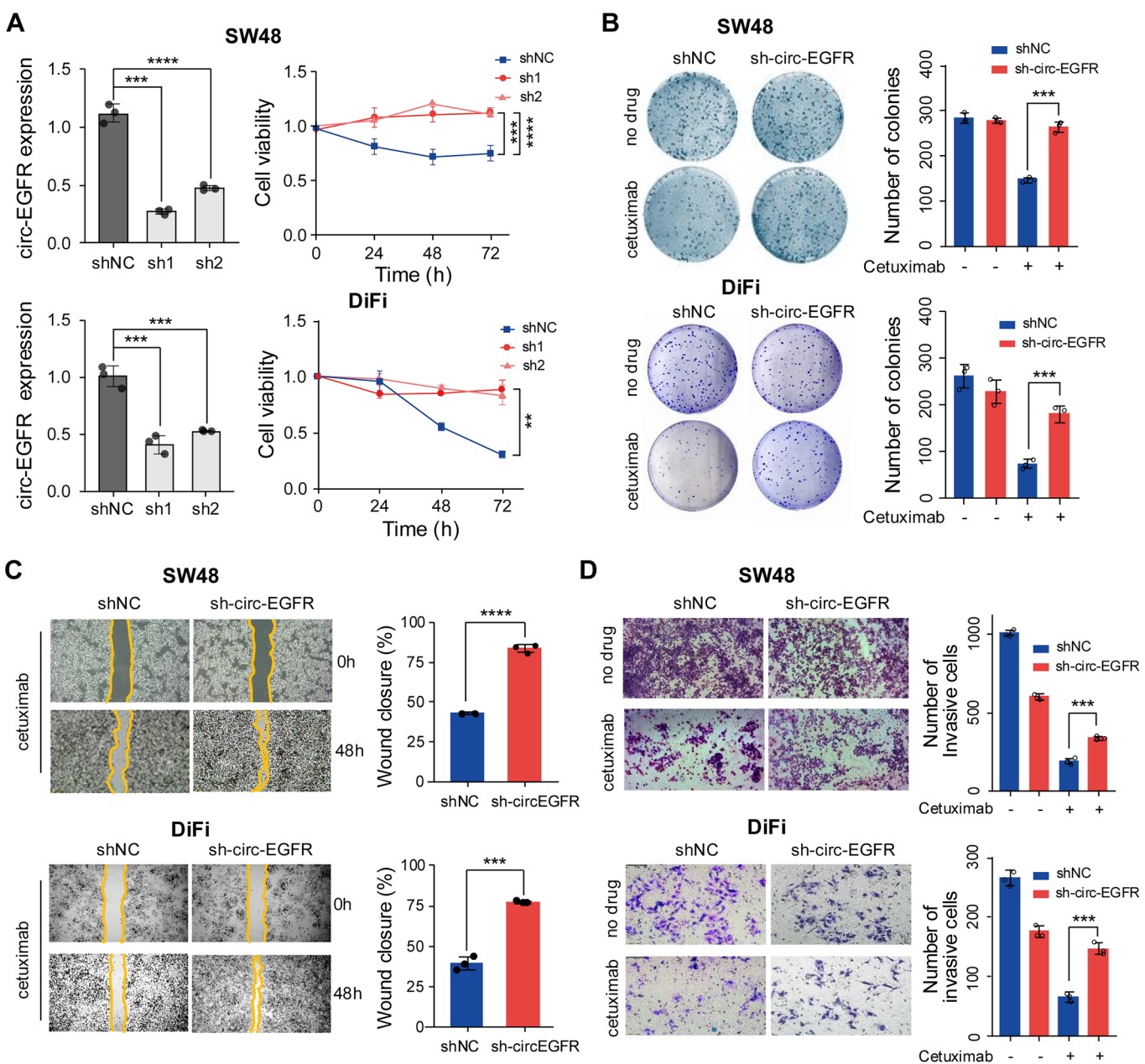

**Figure EV3.** Circ-EGFR depletion inhibits the response of CRC cells to cetuximab in vitro.

(A) The expression of circ-EGFR in stable circ-EGFR-silenced (sh1 and sh2) or negative control (shNC) from DiFi and SW48 cell lines (left). Assessment of cell proliferation capacity by MTT assay in circ-EGFR knockdown (sh1 and sh2) or shNC group from DiFi and SW48 cell lines after 48 h treatment with cetuximab (right). circ-EGFR expression of shNC vs. sh1 in SW48: $p < 0.0001$, shNC vs. sh2 in SW48: $p = 0.0002$, shNC vs. sh1 in DiFi: $p = 0.001$, shNC vs. sh2 in DiFi: $p = 0.0007$ (student's t-test). Cell viability of shNC vs. sh1 in SW48: $p = 0.0004$, shNC vs. sh2 in SW48: $p < 0.0001$, shNC vs. sh1 in DiFi: $p = 0.0004$, shNC vs. sh2 in DiFi: $p = 0.0004$ (two-way ANOVA). (B) Proliferation in stable circ-EGFR knockdown (sh-circ-EGFR) or shNC from DiFi and SW48 cells as determined by colony formation assay. SW48 $p = 0.0001$, DiFi $p = 0.0007$ (student's t-test). (C) Wound healing assay of sh-circ-EGFR or shNC in DiFi and SW48 cell lines after 48 h treatment with cetuximab. Scale bar = 50 μm. SW48 $p < 0.0001$, DiFi $p < 0.0001$ (Student's t-test). (D) Invasion assay of sh-circ-EGFR or shNC in DiFi and SW48 cell lines with or without cetuximab treatment. Scale bar = 100 μm. The number of invading cells was counted in randomly selected three fields. SW48 $p = 0.0002$, DiFi $p = 0.0007$ (student's t-test). Data were representative of at least three independent biological replicates. Bars represent the mean, and error bars indicate SD. *$p < 0.05$, **$p < 0.01$, ***$p < 0.001$, **** $p < 0.0001$, ns not significant.

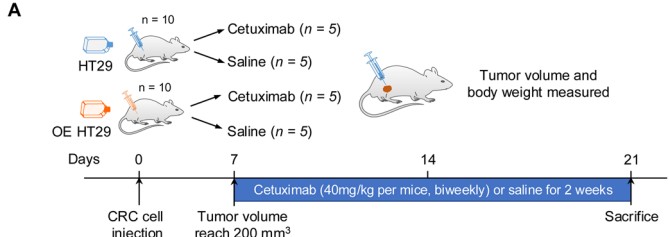

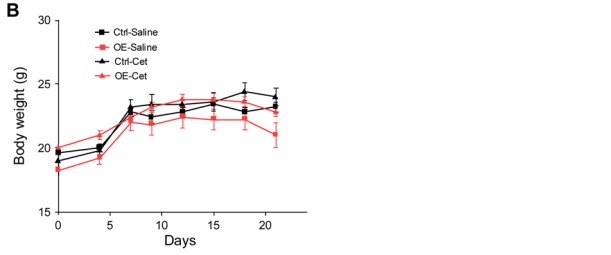

**Figure EV4.   Circ-EGFR enhances the efficacy of cetuximab in vivo.**

(**A**) Schematic diagram of the HT29 cell (transfected with circ-EGFR and mock plasmids)-derived xenograft model in 20 nude mice and the treatment schedule of cetuximab. (**B**) The body weight of nude mice injected with circ-EGFR and mock plasmids in each treatment group was measured at different time points after inoculation. Ctrl Cet: mice inoculated with the HT29 cell line transfected with vector control and treated with cetuximab, Ctrl Saline: mice inoculated with the HT29 cell line transfected with vector control and treated with saline, OE Cet: mice inoculated with circ-EGFR overexpressing HT29 cells and treated with cetuximab, OE Saline: mice inoculated with circ-EGFR overexpressing HT29 cells and treated with saline. Data were representative of five biological replicates. Bars represent the mean, and error bars indicate SD.

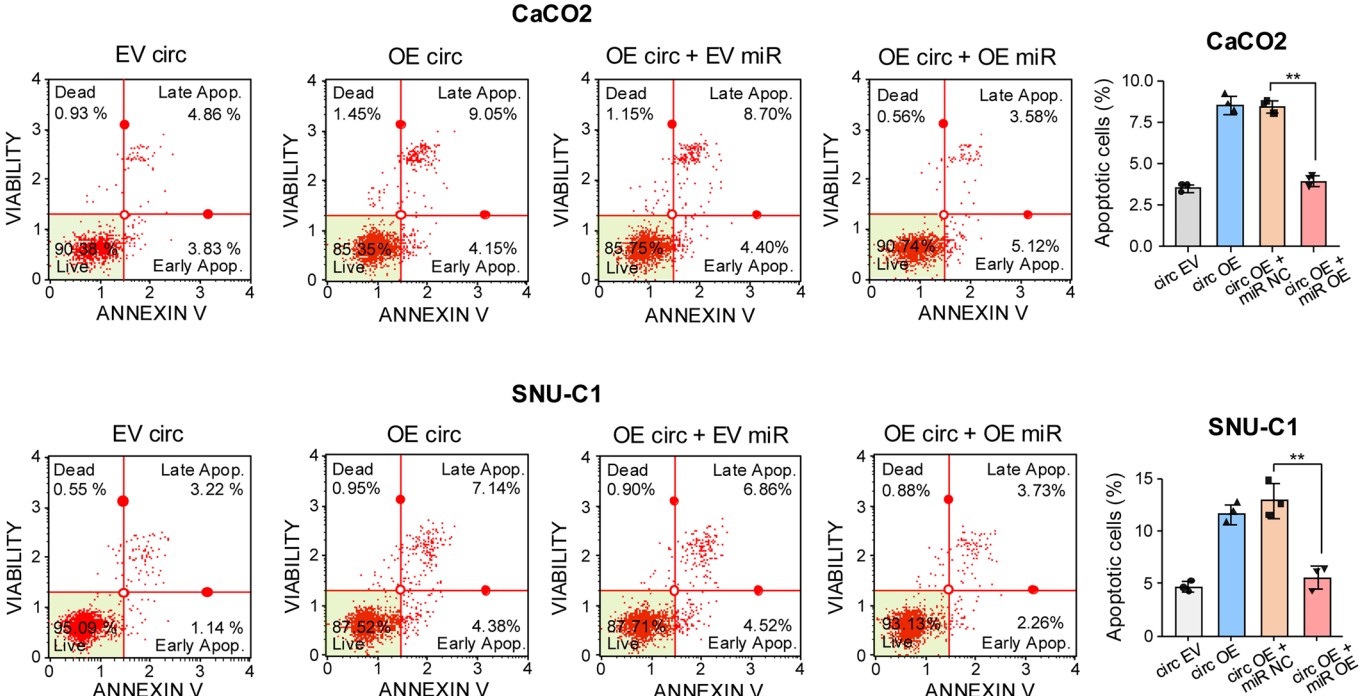

**Figure EV5.   Representative images of cells undergoing apoptosis that stained for the annexin V assay in CaCO2 and SNU-C1 transfected with circ-EGFR and/or miR-942-3p plasmids.**

DiFi $p = 0.006$, SW48 $p = 0.005$ (student's *t*-test). Data were representative of three independent biological replicates. Bars represent the mean, and error bars indicate SD. **$p < 0.01$.

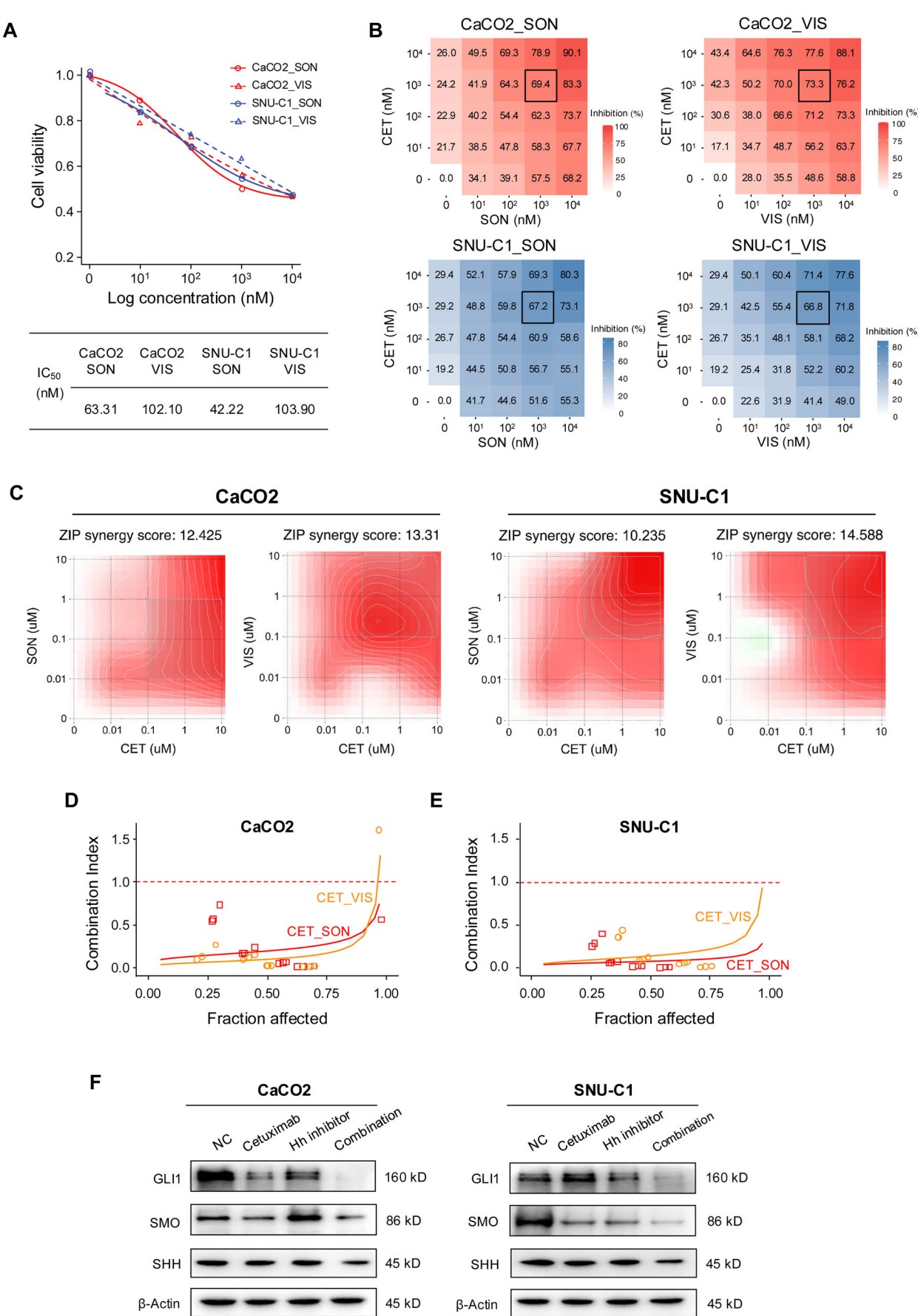

◀ **Figure EV6.  Synergistic effects of cetuximab and hedgehog inhibitor combinations in CRC cells.**

(A) Dose-response curves for Sonidegib or Vismodegib in CaCO2 and SNU-C1 cells. (B) Dose-response matrix (inhibition) for Cetuximab and Sonidegib, or Cetuximab and Vismodegib in CaCO2 and SNU-C1 cells. (C) Synergy distribution and scores for different drug combinations in CaCO2 and SNU-C1 cells were assessed using SynergyFinder with the ZIP model. (D, E) Combination index (CI-Fa) plots for CaCO2 and SNU-C1 cells treated with combinations of Cetuximab and Sonidegib or Cetuximab and Vismodegib. (F) Western blot analysis of Hh signaling pathway components in CaCO2 and SNU-C1 following Cetuximab, Sonidegib or Vismodegib, and the combination therapy. β-Actin served as an internal control.

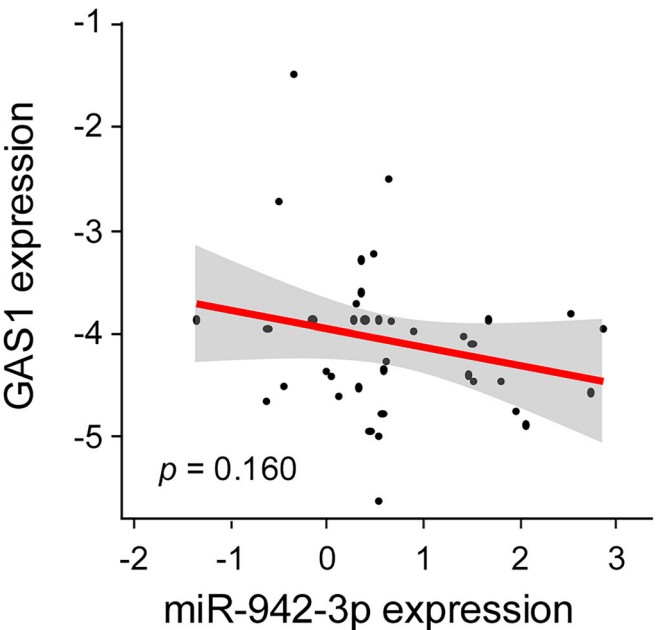

**Figure EV7.  The correlation between miR-942-3p expression and GAS1 expression in the clinical trial cohort.**

$p = 0.160$ (Pearson correlation analysis).

