## [Peer Review File · EMBO Molecular Medicine]

circ-EGFR is a predictor of response to Cetuximab and a potential target in colorectal cancer

Silei Sui, Yuan Li, Joan Maurel, and Ajay Goel

Corresponding author: Ajay Goel (ajgoel@coh.org)

Review Timeline:

Submission Date:	23rd Jan 25
Editorial Decision:	26th Feb 25
Revision Received:	20th Aug 25
Editorial Decision:	12th Sep 25
Revision Received:	21st Oct 25
Accepted:	22nd Oct 25

Editor: Lise Roth

Transaction Report:

26th Feb 2025

Dear Prof. Goel,

Thank you for the submission of your manuscript to EMBO Molecular Medicine, and please accept my apologies for the delay in getting back to you as we were waiting for one referee report. Unfortunately, referee #3 has not yet gotten back to us, but given that both referees #1 and #2 provide similar recommendations, we prefer to make a decision now in order to avoid further delay in the process. Should referee #3 provide a report, we will send it to you, with the understanding that we will not ask you for extensive experiments in addition to the ones required in the enclosed reports from referees #1 and #2. As you will see from the reports below, the referees acknowledge the interest of the study and are overall supporting publication of your work pending appropriate revisions.

Addressing the reviewers' concerns in full will be necessary for further considering the manuscript in our journal, and acceptance of the manuscript will entail a second round of review. EMBO Molecular Medicine encourages a single round of revision only and therefore, acceptance or rejection of the manuscript will depend on the completeness of your responses included in the next, final version of the manuscript. For this reason, and to save you from any frustrations in the end, I would strongly advise against returning an incomplete revision.

We are expecting your revised manuscript within three to four months, if you anticipate any delay, please contact us.

We require:

4) A .docx formatted letter INCLUDING the reviewers' reports and your detailed point-by-point responses to their comments. As part of the EMBO Press transparent editorial process, the point-by-point response is part of the Review Process File (RPF), which will be published alongside your paper.

5) A complete author checklist, which you can download from our author guidelines (<https://www.embopress.org/page/journal/17574684/authorguide#submissionofrevisions>). Please insert information in the checklist that is also reflected in the manuscript. The completed author checklist will also be part of the RPF.

6) All Materials and Methods need to be described in the main text using our 'Structured Methods' format. According to this format, the Methods section includes a Reagents and Tools Table (listing key reagents, experimental models, software and relevant equipment and including their sources and relevant identifiers) followed by a Methods and Protocols section describing the methods, ideally using a step-by-step protocol format. The aim is to facilitate adoption of the methodologies across labs. Please download and fill our Reagents and Tools Table template (.docx), which you can find in our author guidelines: <https://www.embopress.org/page/journal/14693178/authorguide#structuredmethods>.

<https://www.embopress.org/doi/10.15252/msb.20178071>

7) Please note that all corresponding authors are required to supply an ORCID ID for their name upon submission of a revised manuscript.

8) It is mandatory to include a 'Data Availability' section after the Materials and Methods. Before submitting your revision, primary

datasets produced in this study need to be deposited in an appropriate public database, and the accession numbers and database listed under 'Data Availability'. Please remember to provide a reviewer password if the datasets are not yet public (see <https://www.embopress.org/page/journal/17574684/authorguide#dataavailability>).

9) For data quantification: please specify the name of the statistical test used to generate error bars and P values, the number (n) of independent experiments (specify technical or biological replicates) underlying each data point and the test used to calculate p-values in each figure legend. The figure legends should contain a basic description of n, P and the test applied. Graphs must include a description of the bars and the error bars (s.d., s.e.m.). Please provide exact p values.

10) Our journal encourages inclusion of *data citations in the reference list* to directly cite datasets that were re-used and obtained from public databases. Data citations in the article text are distinct from normal bibliographical citations and should directly link to the database records from which the data can be accessed. In the main text, data citations are formatted as follows: "Data ref: Smith et al, 2001" or "Data ref: NCBI Sequence Read Archive PRJNA342805, 2017". In the Reference list, data citations must be labeled with "[DATASET]". A data reference must provide the database name, accession number/identifiers and a resolvable link to the landing page from which the data can be accessed at the end of the reference. Further instructions are available at .

11) We replaced Supplementary Information with Expanded View (EV) Figures and Tables that are collapsible/expandable online. EV Figures should be cited as 'Figure EV1, Figure EV2' etc... in the text and their respective legends should be included in the main text after the legends of regular figures.

12) The paper explained: EMBO Molecular Medicine articles are accompanied by a summary of the articles to emphasize the major findings in the paper and their medical implications for the non-specialist reader. Please provide a draft summary of your article highlighting

13) Author contributions: CRedit has replaced the traditional author contributions section because it offers a systematic machine readable author contributions format that allows for more effective research assessment. Please remove the Authors Contributions from the manuscript and use the free text boxes beneath each contributing author's name in our system to add specific details on the author's contribution. More information is available in our guide to authors.

Please also suggest a visual abstract to illustrate your article as a PNG file 550 px wide x 300-600 px high. A cropped portion of this image will serve as thumbnail for the table of content on our webpage.

16) As part of the EMBO Publications transparent editorial process initiative (see our Editorial at <http://embomolmed.embopress.org/content/2/9/329>), EMBO Molecular Medicine will publish online a Review Process File (RPF) to accompany accepted manuscripts.

In the event of acceptance, this file will be published in conjunction with your paper and will include the anonymous referee

reports, your point-by-point response and all pertinent correspondence relating to the manuscript. Let us know whether you agree with the publication of the RPF and as here, if you want to remove or not any figures from it prior to publication. Please note that the Authors checklist will be published at the end of the RPF.

I look forward to receiving your revised manuscript.

Yours sincerely,

Lise Roth

***** Reviewer's comments *****

Referee #1 (Comments on Novelty/Model System for Author):

Poor description of the material and methods and you be more detailed in teh mani manuscript for the characterisation of circEGFR

Referee #1 (Remarks for Author):

This study identifies circ-EGFR as a predictive biomarker for cetuximab response in metastatic colorectal cancer (mCRC). Patients with high circ-EGFR expression exhibited greater sensitivity to cetuximab, with an AUC of 76.8% in tissue-based analyses and 76.9% in liquid biopsy assessments. Mechanistically, circ-EGFR acts as a molecular sponge for miR-942-3p, leading to GAS1 upregulation and Hedgehog signaling pathway activation, which enhances drug response. Functional studies demonstrated that circ-EGFR overexpression increases cetuximab sensitivity, while its depletion induces resistance in colorectal cancer cells. These findings were further validated in animal models, where tumors with high circ-EGFR expression exhibited greater response to cetuximab treatment. Additionally, the study developed a liquid biopsy test, demonstrating that circ-EGFR can serve as a non-invasive biomarker for predicting therapeutic response. Notably, circ-EGFR-based stratification reduced overtreatment rates from 44.4% to 25%, improving treatment precision. Kaplan-Meier survival analysis indicated that patients with low circ-EGFR expression had significantly shorter progression-free survival (PFS). Furthermore, circ-EGFR levels correlated positively with maximum tumor shrinkage (MTS) post-treatment, underscoring its potential as a clinical tool for personalized mCRC therapy.

This is a well-conducted study, but the manuscript could benefit from improvements in clarity and methodological rigor:

Clarification of Methodology: The section describing how circ-EGFR was detected is insufficiently detailed even when reading the supplementary method. The authors should provide a more precise explanation of their methodology, including specific assays, controls, and validation techniques, to ensure reproducibility.

Discrepancy in circ-EGFR Expression Data: There is a notable discrepancy between the number of patients with low circ-EGFR expression in tissue samples (12/45) and liquid biopsies (72/111). The authors should address this difference by either providing an explanation or speculating on possible reasons for this variation. Differences in sample collection, detection sensitivity, or patient selection criteria could contribute to this discrepancy and should be discussed.

The authors should also discuss other microRNAs that have been implicated in the response to anti-EGFR therapy. Notably, the senior authors of this study previously demonstrated the role of miR-31-3p in cetuximab response through a meta-analysis,

highlighting its potential as a predictive biomarker. Integrating these findings into the discussion would provide a broader context for the role of microRNAs in anti-EGFR treatment resistance and response.

Referee #2 (Comments on Novelty/Model System for Author):

In my opinion this is an interesting manuscript with potential clinical application. However, using a metastatic colorectal cancer (CRC) mouse model or a model that better represents CRC progression compared to subcutaneous xenografts would be welcome. This is important since cetuximab is applied for metastatic CRC patients. In addition, testing the hedgehog pathway directly e.g. by inhibitors is also needed. Do these inhibitors have a similar phenotype compared to circ-EGFR?

Referee #2 (Remarks for Author):

Cetuximab, an EGF receptor targeting monoclonal antibody provides benefit only for a subgroup of patients with metastatic colorectal cancer (mCRC) and wild type KRAS. Thus, many patients receive cetuximab treatment with no clinical efficacy. This results in side effects and unnecessary costs in medical health system. In this manuscript, Sui S et al focus on the role of circ-EGFR in EGFR-targeted therapy in mCRC. By using bioinformatical analysis, in vivo xenograft experiments, molecular biological methods, and overexpression studies, they provide evidence that circ-EGFR functions as a sponge for miR-942-3p, leading to the upregulation of GAS1 and Hedgehog signalling intensity. Of note, circ-EGFR may function as a predictive biomarker for cetuximab response that can also be tested in liquid biopsies. I think this work is interesting scientifically with potential clinical application.

Major concerns:

1. The authors used traditional xenograft in vivo experiments in immunocompromised mice. Since cetuximab is applied in mCRC, it would be important to carry out these studies in a metastatic mouse CRC model, or at least in a model that better represents CRC development compared to subcutaneous xenografts.
2. The authors tested some hedgehog pathway components to prove the mechanism of circ-EGFR. However, this link is still weak, and the functional verification of the sonic hedgehog pathway would be critical e.g. by using inhibitors in in vitro or in vivo experiments.
3. How did the authors define patient groups with high and low circ-EGFR expression? Did the authors carry out an optimization to determine the best cutoff value when classifying patients? This may critically influence clinical results.

Minor concerns:

4. Page 7: „to a reduced overtreatment ratio of 25.0%" -however, Fig 1H has 17.8%.
5. The correlation between circ-EGFR and miR-942-3p is not significant (Fig 6C), thus, the statement on page 12 about the negative correlation is the over-interpretation of the results. Similarly, Fig 6D and Fig 7D do not show convincing correlations, even if it was possible to fit a line on the data points. Could the authors comment on this in the manuscript? What do they think about the clinical consequence of this weak correlation?
6. The discussion lacks mentioning the hedgehog pathway. In addition, further data from the literature would be welcome on the role of GAS and circ-EGFR e.g. in other tumor types.
7. Supplementary Methods, page 3: „miR-16-5p was selected as the optimal control miRNA, based on its stable expression". Could the authors present data on the distribution of the miRNA levels among samples? I think normalization is critical when analyzing miRNAs.
8. Fig EV5 presents apoptosis in CaCO2 and SNU-C1 cell lines. However, this figure is not mentioned and described in the manuscript text. In addition, I guess the EV5 reference on page 12 should be EV6.
9. Could the authors include the description of the statistical methods into the main manuscript text, and not into the supplemental text?

circ-EGFR is a predictor of response to Cetuximab and a potential target in colorectal cancer

Referee #1 (Remarks for Author):

This study identifies circ-EGFR as a predictive biomarker for cetuximab response in metastatic colorectal cancer (mCRC). Patients with high circ-EGFR expression exhibited greater sensitivity to cetuximab, with an AUC of 76.8% in tissue-based analyses and 76.9% in liquid biopsy assessments. Mechanistically, circ-EGFR acts as a molecular sponge for miR-942-3p, leading to GAS1 upregulation and Hedgehog signaling pathway activation, which enhances drug response. Functional studies demonstrated that circ-EGFR overexpression increases cetuximab sensitivity, while its depletion induces resistance in colorectal cancer cells. These findings were further validated in animal models, where tumors with high circ-EGFR expression exhibited greater response to cetuximab treatment. Additionally, the study developed a liquid biopsy test, demonstrating that circ-EGFR can serve as a non-invasive biomarker for predicting therapeutic response. Notably, circ-EGFR-based stratification reduced overtreatment rates from 44.4% to 25%, improving treatment precision. Kaplan-Meier survival analysis indicated that patients with low circ-EGFR expression had significantly shorter progression-free survival (PFS). Furthermore, circ-EGFR levels correlated positively with maximum tumor shrinkage (MTS) post-treatment, underscoring its potential as a clinical tool for personalized mCRC therapy.

This is a well-conducted study, but the manuscript could benefit from improvements in clarity and methodological rigor:

1. **Clarification of Methodology:** The section describing how circ-EGFR was detected is insufficiently detailed, even when reading the supplementary method. The authors should provide a more precise explanation of their methodology, including specific assays, controls, and validation techniques, to ensure reproducibility.

Response: *We appreciate the reviewer's suggestion. We have included a detailed*

explanation of the circ-EGFR detection methodology in the qRT-PCR section of the supplementary methods (**Supp. Data File, Page 3**).

2. Discrepancy in circ-EGFR Expression Data: There is a notable discrepancy between the number of patients with low circ-EGFR expression in tissue samples (12/45) and liquid biopsies (72/111). The authors should address this difference by either providing an explanation or speculating on possible reasons for this variation. Differences in sample collection, detection sensitivity, or patient selection criteria could contribute to this discrepancy and should be discussed.

Response: *We appreciate the reviewer's valuable suggestion. The discrepancy in the number of patients with low circ-EGFR expression between tissue samples and liquid biopsies may be attributed to differences in sample origin, cohort size, and the distribution of clinical responses. (1) Tissue samples primarily reflect tumor-localized circ-EGFR expression, whereas liquid biopsies capture circulating circ-EGFR. These fundamental differences in sample origin are likely the primary contributors to the discrepancies in the distribution of circ-EGFR expression. (2) We used the Youden index to determine optimal cutoff values for each sample type. However, due to the relatively small sample size in the tissue samples, this cutoff may be less stable, which could potentially contribute to the observed discrepancies. (3) Differences in the number of responders and non-responders between the tissue and blood cohorts may have also contributed to this variation. We have included this discussion in the revised submission (see page 17, lines 484–487; note that page numbers in the file with track changes may vary slightly in the clean, formatted document).*

3. The authors should also discuss other microRNAs that have been implicated in the response to anti-EGFR therapy. Notably, the senior authors of this study previously demonstrated the role of miR-31-3p in cetuximab response through a meta-analysis, highlighting its potential as a predictive biomarker. Integrating these findings into the discussion would provide a broader context for the role of microRNAs in anti-EGFR treatment resistance and response.

Response: *Thank you for your question. We have added content on microRNAs in*

*anti-EGFR treatment resistance and response in the discussion section. Please refer to the revised manuscript, **specifically page 16, lines 448–454** (note that page numbers in the file with track changes may vary slightly in the clean, formatted document).*

Referee #2 (Comments on Novelty/Model System for Author):

In my opinion this is an interesting manuscript with potential clinical application. However, using a metastatic colorectal cancer (CRC) mouse model or a model that better represents CRC progression compared to subcutaneous xenografts would be welcome. This is important since cetuximab is applied for metastatic CRC patients. In addition, testing the hedgehog pathway directly e.g. by inhibitors is also needed. Do these inhibitors have a similar phenotype compared to circ-EGFR?

Specific comments:

Cetuximab, an EGF receptor targeting monoclonal antibody provides benefit only for a subgroup of patients with metastatic colorectal cancer (mCRC) and wild type KRAS. Thus, many patients receive cetuximab treatment with no clinical efficiency. This results in side effects and unnecessary costs in medical health system. In this manuscript, Sui S et al focus on the role of circ-EGFR in EGFR-targeted therapy in mCRC. By using bioinformatical analysis, in vivo xenograft experiments, molecular biological methods, and overexpression studies, they provide evidence that circ-EGFR functions as a sponge for miR-942-3p, leading to the upregulation of GAS1 and Hedgehog signalling intensity. Of note, circ-EGFR may function as a predictive biomarker for cetuximab response that can also be tested in liquid biopsies. I think this work is interesting scientifically with potential clinical application.

Major concerns:

1. The authors used traditional xenograft in vivo experiments in immunocompromised

mice. Since cetuximab is applied in mCRC, it would be important to carry out these studies in a metastatic mouse CRC model, or at least in a model that better represents CRC development compared to subcutaneous xenografts.

Response: *Thank you very much for your insightful suggestion. We have conducted additional in vivo experiments using a mouse model of liver metastasis to better mimic the metastatic progression of CRC and the clinical application of cetuximab. The results of these experiments have been incorporated into the revised manuscript (Figure 3, see page 9, lines 245–262; note that page numbers in the file with track changes may vary slightly in the clean, formatted document).*

2. The authors tested some hedgehog pathway components to prove the mechanism of circ-EGFR. However, this link is still weak, and the functional verification of the sonic hedgehog pathway would be critical e.g. by using inhibitors in *in vitro* or *in vivo* experiments.

Response: *Thank you for your constructive comment. To further validate the Hedgehog pathway's role in mediating cetuximab response in CRC, we performed in vitro functional experiments using two FDA-approved Hedgehog inhibitors (Sonidegib and Vismodegib) to assess their effects on cetuximab sensitivity. The results have been added to the revised manuscript (see pages 12, lines 344–353, and 13, lines 354–365; note that page numbers in the file with track changes may vary slightly in the clean, formatted document).*

3. How did the authors define patient groups with high and low circ-EGFR expression? Did the authors carry out an optimization to determine the best cutoff value when classifying patients? This may critically influence clinical results.

Response: *We appreciate the reviewer's insightful comment. To define patient groups with high and low circ-EGFR expression, we used separate cutoff values for tissue and blood samples, because circ-EGFR expression may differ between tumor-localized and circulating compartments. To determine the optimal cutoff for each sample type, we used Youden's index to identify the cut-off that maximizes both sensitivity and specificity for classification.*

Minor concerns:

4. Page 7: to a reduced overtreatment ratio of 25.0%" -however, Fig 1H has 17.8%.

Response: *Thank you for bringing that to our attention. It was indeed a typographical error, and we have corrected it accordingly (Page 7, Line 178).*

5. The correlation between circ-EGFR and miR-942-3p is not significant (Fig 6C), thus, the statement on page 12 about the negative correlation is the over-interpretation of the results. Similarly, Fig 6D and Fig 7D do not show convincing correlations, even if it was possible to fit a line on the data points. Could the authors comment on this in the manuscript? What do they think about the clinical consequence of this weak correlation?

Response: *We appreciate the reviewer's insightful comment. We have revised the original statement in the manuscript accordingly (see Page 13, Lines 373–376). We acknowledge that the observed correlations are not robust. These findings likely reflect the inherent complexity of biological systems, particularly in clinical specimens where inter-patient heterogeneity, post-transcriptional regulation, tumor microenvironmental factors, and sampling variability can weaken detectable associations. Additionally, the limited sample size may have further contributed to the lack of statistical robustness.*

6. The discussion lacks mentioning the hedgehog pathway. In addition, further data from the literature would be welcome on the role of GAS1 and circ-EGFR e.g. in other tumor types.

Response: *Thank you for your question. We have incorporated additional content on the roles of GAS1 and circ-EGFR in the discussion section. Please refer to the revised manuscript, Page 16, lines 462–466 (note that page numbers in the file with track changes may vary slightly in the clean, formatted document).*

7. Supplementary Methods, page 3: miR-16-5p was selected as the optimal control miRNA, based on its stable expression". Could the authors present data on the distribution of the miRNA levels among samples? I think normalization is critical when analyzing miRNAs.

Response: *The expression levels of miRNAs in cell lines and tissues are provided in the raw data and will be uploaded for reference.*

8. Fig EV5 presents apoptosis in CaCO2 and SNU-C1 cell lines. However, this figure is not mentioned and described in the manuscript text. In addition, I guess the EV5 reference on page 12 should be EV6.

Response: *Thank you for bringing this to our attention. We have revised the manuscript accordingly, ensuring that Figures EV5 and EV6 are correctly referenced and described.*

9. Could the authors include the description of the statistical methods into the main manuscript text, and not into the supplemental text?

Response: *Thank you for your suggestion. We have revised the manuscript accordingly and relocated the description of the statistical methods from the supplemental text to the main manuscript (**Page 19, Lines 545–556**; note that page numbers in the file with track changes may vary slightly in the clean, formatted document).*

12th Sep 2025

Dear Prof. Goel,

Thank you for submitting your revised study. We have now received the feedback from the referees, and as you will see below, they are satisfied with the revisions. I will therefore be able to accept your manuscript once the following editorial concerns are addressed:

1/ Manuscript text:

- Please indicate in track changes mode any new modification in the text.
- "Materials and Methods" should be renamed "Methods". The supplementary methods should be added to the main manuscript text.
 - o Patient specimens: Please provide the full statement confirming that the experiments conformed to the principles set out in the WMA Declaration of Helsinki and the Department of Health and Human Services Belmont Report. State details of authority granting ethics approval (IRB or equivalent committee(s), provide reference number for approval. Report the clinical trial registration number (at ClinicalTrials.gov or equivalent), where applicable. If collected and within the bounds of privacy constraints report on age, sex and gender or ethnicity for all study participants.
 - o Cell cultures: please indicate whether the cells were authenticated and tested for mycoplasma contamination.
 - o Animals: please provide housing and husbandry conditions.
 - o Antibodies: please provide dilutions/concentrations.
 - o Statistics: please provide a statement on sample size, inclusion/exclusion criteria, blinding and randomization.
- The Availability of data and materials section should be removed.
- The Ethical Approval section should be moved to the Methods.
- The list of abbreviations should be removed.
- Funding should be merged with Acknowledgements.
- Data availability section: Please remove "All data supporting the findings of this study are available within the article and its Expanded View files." This section should list large primary datasets that are part of this study (and newly generated for this study). In case you have no data that requires deposition in a public database, please state so in this section ("This study includes no data deposited in external repositories.").

2/ Figures:

- Please provide individual data points in your graphs.
- The suppl. figures should be uploaded as individual high resolution figure files, and their legends should be in the main manuscript text, after the main figure legends and under the heading "Expanded View Figure Legends".
- Please remove Figure EV7 from the manuscript text - it is different from the Figure EV7 that is uploaded in the suppl. figures document, please check.
- The tables EV1 - EV4 should be uploaded as individual files.
- Please address the queries from our data editors in the figure legends:
 1. Please define the annotated p values ****/****/**/* as well as provide the exact p-values for the same in the legend of figure 2A, B, C, D, E, F; 3A, 4A, C, E, F, H; 5A-C; 6E, 7A, B; 8A, E; EV5 as appropriate.
 2. Please note that the exact p values are not provided in the legends of figures 1A, D, F, H, I; 7C, D; 8C, D; EV1; EV3 A-D
 3. Please indicate the statistical test used for data analysis in the legends of figures 2A, B, C, D, E, F; 3A; 4A, C, E, F, H; 5A-C; 6B, E; 7A, B, C, D; 8A, C, D, E; EV1; EV3 A-D; EV5, EV7.
 4. Please note that the box plots need to be defined in terms of minima, maxima, centre, bounds of box and whiskers, and percentile in the legends of figures 1F, 7A, B; 8A
 5. Please note that information related to n is missing in the legends of figures 3A, 8A, E; EV1; EV4 B, EV5
 6. Please note that the error bars are not defined in the legends of figures 1A, D, H; 2A, B, C, D, E, F; 4A, C, E, F, H; 5A-C; 6B, E; 8E; EV1; EV3 A-D; EV4 B, EV5
 7. Please note that for heatmap present in figure EV6 B a numbered scale bar is not provided. This needs to be rectified.

3/ Thank you for providing Source Data:

- Fig. 2D: please provide fcs files.
- Fig 5C, 6C, D, F: please provide jpg, tiff or PDF format.
- Fig. 5C: numerical data are missing for SNU-C1.
- in general, please provide individual data points (not mean).

4/ Checklist:

- please fill in Cell materials/authentication and mycoplasma contamination.
- please fill in Experimental animals/housing and husbandry conditions.
- please fill in Experimental study design and statistics/ sample size and blinding.
- please fill in Ethics/ human participants (2 upper subsections). Please check the subsection "Specimen and field samples" as I

don't think it applies to your study.

- Data Availability: please check whether you need to fill the subsections on primary datasets.

5/ The paper explained:

I have edited the last paragraph of The Paper Explained to match our style and format, please let me know if you agree or amend as you see fit:

"Impact: By elucidating the role of circ-EGFR in modulating tumor response to cetuximab via the circ-EGFR/miR-942-3p/GAS1/Hh regulatory axis, our findings provide key molecular insights into the biologic changes associated with varying response to anti-EGFR therapy in CRC. The successful implementation of the circ-EGFR predictive approach is anticipated to inform patient selection, enhance personalized management strategies, and ultimately improve patient outcomes."

6/ I added minor modifications to your synopsis text, please let me know if you agree or amend as you see fit:

"Circ-EGFR was identified as a novel predictive biomarker for cetuximab efficacy in KRAS wild-type metastatic colorectal cancer, which was successfully translated into a non-invasive liquid biopsy assay for predicting responses to anti-EGFR therapy.

- Tissue-based circ-EGFR biomarker was shown to effectively stratify cetuximab responders from non-responders in mCRC, with an AUC of 76.8%.

- The therapeutic efficacy of cetuximab was potentiated by circ-EGFR in both in vitro and in vivo CRC models.

- Mechanistically, circ-EGFR functions as a sponge for miR-942-3p and upregulates GAS1, which activates the Hedgehog pathway.

- A plasma-based circ-EGFR assay was developed and validated as a non-invasive predictor of anti-EGFR therapy response (AUC: 76.9%)."

Thank you for providing a nice visual abstract. Please upload it as a jpeg/tiff/png file 550 px wide x 300-600 px high, and make sure that the text remains legible. A cropped portion of this image will serve as thumbnail for the table of content on our webpage.

7/ As part of the EMBO Publications transparent editorial process initiative (see our Editorial at <http://embomolmed.embopress.org/content/2/9/329>), EMBO Molecular Medicine will publish online a Review Process File (RPF) to accompany accepted manuscripts.

This file will be published in conjunction with your paper and will include the anonymous referee reports, your point-by-point response and all pertinent correspondence relating to the manuscript. Let us know whether you agree with the publication of the RPF and as here, if you want to remove or not any figures from it prior to publication.

I look forward to receiving your revised manuscript.

Yours sincerely,

Lise Roth

***** Reviewer's comments *****

Referee #1 (Comments on Novelty/Model System for Author):

I am pleased to review this revised manuscript. The authors have significantly improved the quality of their work, and their responses to my major comments are thorough and greatly appreciated.

Referee #1 (Remarks for Author):

I thank the authors to their answers who have significantly improved the quality of their work

Referee #2 (Comments on Novelty/Model System for Author):

In the revised manuscript, the authors involved a liver metastatic model, and they also tested hedgehog pathway inhibitors with cetuximab. These two were critical to validate their findings in more appropriate systems.

Referee #2 (Remarks for Author):

Many thanks to the authors for their corrections and the novel experiments, they successfully answered my concerns.

The authors addressed the remaining editorial issues.

22nd Oct 2025

Dear Prof. Goel,

Thank you for your patience and for submitting your revised files. I am pleased to inform you that your manuscript is accepted for publication and is now being sent to our publisher to be included in the next available issue of EMBO Molecular Medicine!

Yours sincerely,

Lise Roth
